# SE(3)-equivariant prediction of molecular wavefunctions and electronic densities

**Oliver T. Unke**[1,2,8,*,†]
oliver.unke@gmail.com

**Mihail Bogojeski**[1,8,*]
mihailbogojeski@gmail.com

**Michael Gastegger**[1,2,9]       **Mario Geiger**[3]       **Tess Smidt**[4,5]

**Klaus-Robert Müller** [1,6,7,8,10]
klaus-robert.mueller@tu-berlin.de

[1] Machine Learning Group, Technische Universität Berlin, 10587 Berlin, Germany
[2] DFG Cluster of Excellence "Unifying Systems in Catalysis" (UniSysCat), Technische Universität Berlin, 10623 Berlin, Germany
[3] Institute of Physics, École Polytechnique Fédérale de Lausanne, 1015 Lausanne, Switzerland
[4] Computational Research Division, Lawrence Berkeley National Laboratory, Berkeley, CA 94720
[5] Center for Advanced Mathematics for Energy Research Applications (CAMERA), Lawrence Berkeley National Laboratory, Berkeley, CA 94720
[6] Department of Artificial Intelligence, Korea University, Anam-dong, Seongbuk-gu, Seoul 02841, Korea
[7] Max Planck Institute for Informatics, Stuhlsatzenhausweg, 66123 Saarbrücken, Germany
[8] BIFOLD – Berlin Institute for the Foundations of Learning and Data, Berlin, Germany
[9] BASLEARN – TU Berlin/BASF Joint Lab for Machine Learning, Technische Universität Berlin, 10587 Berlin, Germany
[10] Google Research, Brain Team, Berlin, Germany

## Abstract

Machine learning has enabled the prediction of quantum chemical properties with high accuracy and efficiency, allowing to bypass computationally costly *ab initio* calculations. Instead of training on a fixed set of properties, more recent approaches attempt to learn the electronic wavefunction (or density) as a central quantity of atomistic systems, from which all other observables can be derived. This is complicated by the fact that wavefunctions transform non-trivially under molecular rotations, which makes them a challenging prediction target. To solve this issue, we introduce general SE(3)-equivariant operations and building blocks for constructing deep learning architectures for geometric point cloud data and apply them to reconstruct wavefunctions of atomistic systems with unprecedented accuracy. Our model achieves speedups of over three orders of magnitude compared to *ab initio* methods and reduces prediction errors by up to two orders of magnitude compared to the previous state-of-the-art. This accuracy makes it possible to derive properties such as energies and forces directly from the wavefunction in an end-to-end manner. We demonstrate the potential of our approach in a transfer learning application, where a model trained on low accuracy reference wavefunctions implicitly learns to correct for electronic many-body interactions from observables computed at a higher level of theory. Such machine-learned wavefunction surrogates pave the way towards novel semi-empirical methods, offering resolution at an electronic

---
*These authors contributed equally.
†Work done at TU Berlin prior to joining Google Research.

35th Conference on Neural Information Processing Systems (NeurIPS 2021).

level while drastically decreasing computational cost. Additionally, the predicted wavefunctions can serve as initial guess in conventional *ab initio* methods, decreasing the number of iterations required to arrive at a converged solution, thus leading to significant speedups without any loss of accuracy or robustness. While we focus on physics applications in this contribution, the proposed equivariant framework for deep learning on point clouds is promising also beyond, say, in computer vision or graphics.

# 1 Introduction

Machine learning (ML) methods are becoming increasingly popular in quantum chemistry as a means to circumvent expensive *ab initio* calculations, and led to advances in a broad range of applications, including the construction of potential energy surfaces [1–8], prediction of electron densities and density functionals [9–14], and development of models capable of predicting a range of physical observables across chemical space [15–29]. Typically, such models are trained on reference data for a predetermined set of quantum chemical properties and need to be retrained if other properties are required. However, if a model is capable of predicting the wavefunction, expectation values for *any* observable can be derived from it. Unfortunately, such an approach is complicated by the fact that wavefunctions are typically expressed in terms of rotationally equivariant basis functions, introducing non-trivial transformations under molecular rotations, which are difficult to learn from data. To solve this issue, we propose several SE(3)-equivariant operations for deep learning architectures for geometric point cloud data, which capture the effects of translations and rotations without needing to learn them explicitly. We assemble these building blocks to construct PhiSNet, a novel deep learning (DL) architecture for predicting wavefunctions and electronic densities, which is significantly more accurate than non-equivariant models. For the first time, sufficient accuracy is reached to predict properties like energies and forces directly from the wavefunction and in end-to-end manner.

This makes it possible to learn wavefunctions that lead to modified properties, which is interesting from an inverse design perspective; or the development of novel machine-learned semi-empirical methods, for example by learning a correction to the wavefunction that mimics the effects of electron correlation. Such hybrid methods maintain the accuracy and generality of high level electronic structure calculations while drastically reducing their computational cost. In addition, the predicted wavefunctions can serve as initial guess to speed up conventional *ab initio* methods.
Beyond physics, other applications of our proposed equivariant DL architecture to e.g. computer vision or graphics are conceivable – whenever accurate invariant analyses of high dimensional point clouds are of importance.

In summary, this work provides the following contributions:

- We describe general SE(3)-equivariant operations and building blocks for constructing DL architectures for geometric point cloud data.

- We propose PhiSNet, a neural network for predicting wavefunctions and electronic densities from equivariant atomic representations, ensuring physically correct transformation under translations and rotations.

- We apply PhiSNet to predict wavefunctions and electronic densities of several molecules and show that our model reduces prediction errors of electronic structure properties by a factor of up to two orders of magnitude compared to the previous state-of-the-art and achieves speedups of over three orders of magnitude compared to *ab initio* solutions.

- We showcase a novel transfer-learning application, where a model trained on low accuracy wavefunctions is adapted to predict properties computed at a higher level of theory by learning a correction that implicitly captures the effects of many-body electron correlation.

- We demonstrate that the predicted wavefunctions can serve as initial guess in conventional quantum chemistry methods, leading to significant speedups without sacrificing the accuracy or robustness of *ab initio* solutions.

In principle, our method could also be used to construct orbital features as inputs for methods like OrbNet [30], which otherwise rely on semi-empirical or *ab initio* methods.

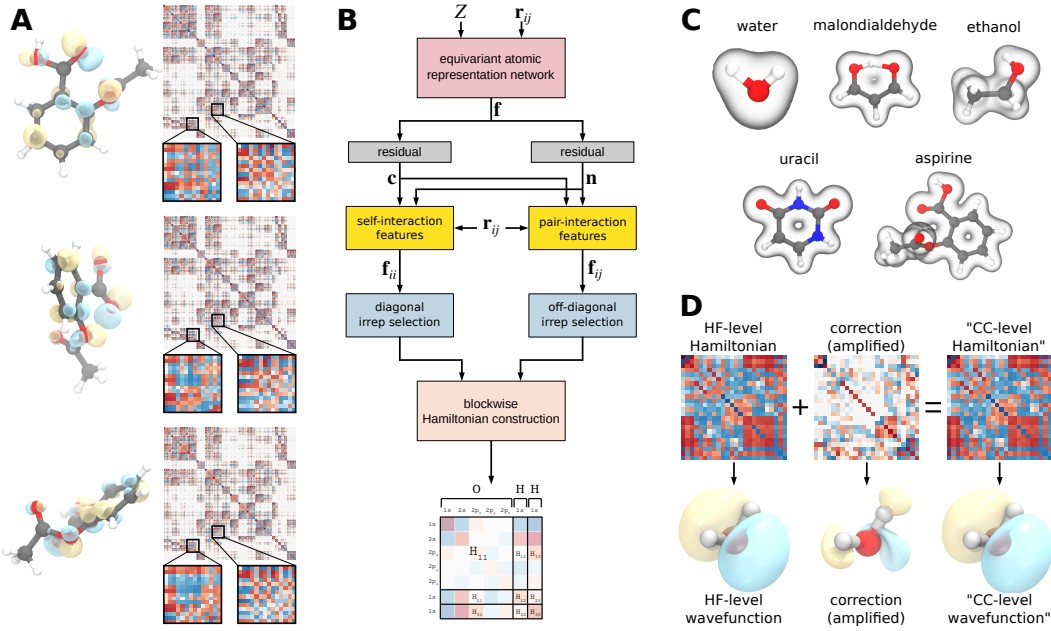

Figure 1: **A**: Illustration of an aspirine molecule and its highest occupied molecular orbital (HOMO) in three different orientations, showing how the wavefunction (left) and Hamiltonian matrix (right) change with respect to rotations. **B**: Overview of the proposed PhiSNet architecture. The atomic representation network creates atom-wise equivariant features, which are used to produce self-interaction and pair-interaction features (Fig. S2), from which the Hamiltonian matrix is constructed block-by-block (Fig. S5). **C**: Visualisation of electronic densities (squared wavefunction) of various molecules predicted with our approach. **D**: Illustration of a transfer learning application, where a model pretrained on Hartree-Fock (HF) Hamiltonians is fine-tuned to match energies and forces derived from highly accurate coupled cluster (CC) calculations. The model achieves this by learning a correction to the Hamiltonian matrix, which mimics the effects of many-body electron correlation. The effective "CC-level" Hamiltonian can be thought of as a HF-level Hamiltonian plus a correction term. The HOMO is shown to visualize subtle changes to the wavefunction (the correction is amplified in magnitude by a factor of $10^3$ for better visibility).

## 2 Related work

Only a small number of studies apply ML to the challenging problem of modeling the wavefunction directly [31]. This is usually done by predicting Hamiltonian matrices, from which the wavefunction can be obtained by solving a generalized eigenvalue problem. The earliest such study we are aware of is by Hegde and Bowen [32], who used kernel ridge regression to learn the Hamiltonian matrix for two simple case studies. Later, Schütt et al. [33] proposed the SchNOrb neural network architecture, which constructs the Hamiltonian matrix of molecules in a block-wise manner from atom-pair features. Recently, Li et al. [34] presented a deep neural network architecture for predicting the Hamiltonian matrix of simple periodic crystals. However, none of these models form their predictions in a rotationally equivariant manner, i.e. they need to learn how to predict the Hamiltonian matrix for all possible orientations of the system of interest, which requires large amounts of training data. Even when data augmentation via random rotations [35] or special Hamiltonian representations [36] are used to mitigate this issue, the final model is only approximately equivariant, i.e. properties derived from the wavefunction can change unphysically when the system is rotated or the frame of reference is changed. Here, we draw upon insights from a range of SE(3)-equivariant models [37–48] to ensure that predictions exactly preserve the physically correct dependence with respect to the orientation of inputs (for a more detailed discussion see the supplement). We would like to remark that after the initial submission of this manuscript, Nigam et al. [49] introduced a complementary method to construct equivariant representations for Hamiltonian matrices, e.g. for the use in kernel machines.

# 3  Background

The aim of most quantum chemistry methods is to solve the electronic Schrödinger equation

$$\hat{H}_{\text{el}}\Psi_{\text{el}} = E_{\text{el}}\Psi_{\text{el}}, \tag{1}$$

where $\hat{H}_{\text{el}}$ is the Hamiltonian operator describing the interactions and motion of the electrons, $\Psi_{\text{el}}$ is the electronic wavefunction and $E_{\text{el}}$ is the ground state energy. After $\Psi_{\text{el}}$ is determined, all physical observables (beyond $E_{\text{el}}$) can be derived by applying other operators (similar to $\hat{H}_{\text{el}}$) to the wavefunction and reading out the corresponding eigenvalues [50]. In practice, Eq. 1 is usually solved by expressing $\Psi_{\text{el}}$ as an antisymmetric product of molecular orbitals

$$\psi_i = \sum_j C_{ij}\phi_j \,, \tag{2}$$

which are written as linear combinations of atom-centered basis functions $\phi$. This leads to the equation

$$\mathbf{HC} = \epsilon\mathbf{SC}\,, \tag{3}$$

where the Hamiltonian is written as a matrix $\mathbf{H}$ with entries $H_{ij} = \int \phi_i^*(\mathbf{r})\hat{H}_{\text{el}}\phi_j(\mathbf{r})d\mathbf{r}$ ($\mathbf{r}$ denotes the electronic coordinates). The overlap matrix $\mathbf{S}$ with entries $S_{ij} = \int \phi_i^*(\mathbf{r})\phi_j(\mathbf{r})d\mathbf{r}$ has to be introduced and a generalized eigenvalue problem must be solved, because the basis functions $\phi$ are typically not orthonormal. The eigenvectors $\mathbf{C}$ specify the wavefunction $\Psi_{\text{el}}$ via the coefficients $C_{ij}$ of the molecular orbitals (Eq. 2) and the eigenvalues $\epsilon$ are the corresponding orbital energies. Since eigenvectors are only defined up to sign changes, predicting $\mathbf{H}$ (instead of $\mathbf{C}$) is preferable for ML applications. This is a challenging task, because the basis functions are typically products of a radial component and spherical harmonics, which introduces non-trivial dependencies of the matrix elements with respect to the orientation of the chemical system (see Fig. 1A). Spherical harmonics $Y_l^m$ of degree $l = 0, \dots, \infty$ and order $m = -l, \dots, l$ form a complete orthonormal basis for functions on the surface of a sphere and can be used to derive irreducible representations (irreps) of the 3D rotation group $\text{SO}(3)$. In other words, they are rotationally equivariant, which means that when $\mathbf{r}$ is rotated, the values of $Y_l^m(\mathbf{r})$ change accordingly. Since $\Psi_{\text{el}}$ is expressed with spherical harmonic basis functions, the entries of $\mathbf{H}$ transform predictably under rotations (a more detailed overview of quantum chemistry fundamentals, groups, equivariance, and spherical harmonics is given in Section A of the supplement).

# 4  Deep learning architecture for molecular wavefunctions

Deep message-passing neural networks (MPNNs) [51] for quantum chemistry applications, such as DTNN [18] or SchNet [19], model physical properties of chemical systems as a sum over atomic contributions predicted from features $\mathbf{x}_i \in \mathbb{R}^F$ for each atom $i$. Starting from initial element-specific embeddings, the features are constructed by iteratively exchanging "messages" between neighboring atoms $i$ and $j$, which depend on the current $\mathbf{x}_i$ and $\mathbf{x}_j$ and their distance $r_{ij}$. Since geometric information enters only in the form of pairwise distances, the final atomic features are rotationally invariant by construction. This is desirable when they are used to predict a quantity that itself is rotationally invariant, for example the potential energy. To predict observables that change under rotation, e.g. electric moments or the electronic Hamiltonian, a natural extension is to instead construct rotationally equivariant features. To see how this can be achieved, it is useful to think of the $F$ entries of atomic feature vectors $\mathbf{x}_i \in \mathbb{R}^F$ as different "channels", where each channel carries scalar information about the chemical environment of atom $i$. To construct rotationally equivariant features, each scalar channel can be replaced by values derived from the spherical harmonics up to a maximum degree $L$, i.e. there are now $F \times (L+1)^2$ entries (each spherical harmonic degree $l \in \{0, \dots, L\}$ contributes $2l + 1$ values for all possible orders $m \in \{-l, \dots, l\}$). Our proposed model, which we call PhiSNet, shares basic design principles with PhysNet [23], but uses equivariant (instead of invariant) operations throughout its architecture. Contrary to most other MPNNs, instead of directly predicting chemical properties from atomic features, PhiSNet constructs the Hamiltonian matrix in a block-wise manner from equivariant representations. All known physical symmetries of the Hamiltonian are preserved by construction, an essential constraint which is not satisfied by existing DL architectures for chemical applications.

**Notation** Whenever equivariant features are discussed, bold symbols (e.g. $\mathbf{x}$) refer to the collection of all $F \times (L+1)^2$ entries for all feature channels $F$ and spherical harmonics degrees $l \in \{0, \ldots, L\}$, whereas a superscript $l$ in parentheses (e.g. $\mathbf{x}^{(l)}$) is used to refer only to the $F \times (2l+1)$ entries of degree $l$. Similarly, $\mathbf{Y}(\mathbf{r})$ refers to the collection of all $1 \times (L+1)^2$ spherical harmonics with distinct combinations of $l$ and $m$, whereas $\mathbf{Y}^{(l)}(\mathbf{r})$ refers to the $1 \times (2l+1)$ values for degree $l$. The notation $\mathbf{c} \circ \mathbf{x}$ denotes a Hadamard product between matrices, i.e. $\mathbf{c} \in \mathbb{R}^{F \times (L+1)^2}$ and $\mathbf{x} \in \mathbb{R}^{F \times (L+1)^2}$ are multiplied entry-wise. When there is no one-to-one correspondence between the entries in $\mathbf{c}$ and $\mathbf{x}$, e.g. when $\mathbf{c} \in \mathbb{R}^F$ and $\mathbf{x} \in \mathbb{R}^{F \times (L+1)^2}$, $\mathbf{c} \circ \mathbf{x}$ implies that $\mathbf{c}$ is "broadcasted" across the missing dimensions, i.e. each of the $(L+1)^2$ "slices" of $\mathbf{x}$ is multiplied entry-wise with $\mathbf{c}$ and the result $\mathbf{c} \circ \mathbf{x}$ has dimensions $F \times (L+1)^2$. Double-struck digits denote an irreducible representation (irrep) with the corresponding number of dimensions $2l+1$. For example, $\mathbb{1}$ refers to a one-dimensional irrep of degree $l = 0$ and $\mathbb{3}$ to a three-dimensional irrep of degree $l = 1$. By abuse of terminology, the term "irrep" is also used for the individual $(2l+1)$-dimensional components along the $F$ feature dimensions of $\mathbf{x} \in \mathbb{R}^{F \times (L+1)^2}$. When a collection of equivariant features $\mathbf{x}^{(l)} \in \mathbb{R}^{F_{\text{in}} \times (2l+1)}$ is multiplied by a matrix $\mathbf{M} \in \mathbb{R}^{F_{\text{out}} \times F_{\text{in}}}$, the result is $\mathbf{M}\mathbf{x}^{(l)} \in \mathbb{R}^{F_{\text{out}} \times (2l+1)}$, i.e. the ordinary rules for matrix multiplication apply.

## 4.1 SE(3)-equivariant neural network building blocks

In the following, we describe general-purpose operations for building SE(3)-equivariant MPNNs, which can also be used outside a chemical context to build feature representations for other point cloud data. Additionally, we discuss necessary modifications to established neural network components (e.g. linear layers or activation functions) for keeping feature representations rotationally equivariant.

**Activation functions** may only be applied to scalar ($l = 0$) features, or else the output loses its equivariant properties:

$$\boldsymbol{\sigma}(\mathbf{x})^{(l)} = \begin{cases} \sigma(\mathbf{x}^{(l)}) & l = 0 \\ \mathbf{x}^{(l)} & l > 0 \end{cases}. \tag{4}$$

Here, $\sigma$ can be any activation function and the notation $\sigma(\mathbf{x}^{(l)})$ means that $\sigma$ is applied to $\mathbf{x}^{(l)}$ entry-wise. In this work, a generalized SiLU [52, 53] activation function (also known as Swish [54]) given by

$$\sigma(x) = \frac{\alpha x}{1 + e^{-\beta x}} \tag{5}$$

is used, where $\alpha$ and $\beta$ are both learned and separate parameters are kept for all feature channels and instances of $\sigma$ [27] (see Section B.1 in the supplement for additional details).

**Linear layers** are applied to each degree $l$ according to

$$\text{linear}_{F_{\text{in}} \to F_{\text{out}}}(\mathbf{x})^{(l)} = \begin{cases} \mathbf{W}_l \mathbf{x}^{(l)} + \mathbf{b} & l = 0 \\ \mathbf{W}_l \mathbf{x}^{(l)} & l > 0 \end{cases}, \tag{6}$$

where $\mathbf{W} \in \mathbb{R}^{F_{\text{out}} \times F_{\text{in}}}$ and $\mathbf{b} \in \mathbb{R}^{F_{\text{out}}}$ are weights and biases, respectively. The subscript $l$ is used to distinguish the weights for different degrees $l$, i.e. separate linear transformations are applied to the features of each degree $l$. The bias term must be omitted for $l > 0$ so that output features stay rotationally equivariant.

**Tensor product contractions** are used to couple two equivariant feature representations $\mathbf{x}^{(l_1)}$ and $\mathbf{y}^{(l_2)}$ to form new features $\mathbf{z}^{(l_3)}$. The (reducible) tensor product $\mathbf{x}^{(l_1)} \otimes \mathbf{y}^{(l_2)}$ of two irreps has $(2l_1+1)(2l_2+1)$ dimensions and can be expanded into a direct sum of irreducible representations, e.g. $\mathbb{3} \otimes \mathbb{5} = \mathbb{3} \oplus \mathbb{5} \oplus \mathbb{7}$. In general, the value for order $m_3$ of the irrep of degree $l_3 \in \{|l_1 - l_2|, \ldots, l_1 + l_2\}$ in the direct sum representation of the tensor product $\mathbf{x}^{(l_1)} \otimes \mathbf{y}^{(l_2)}$ is given by

$$\left(\mathbf{x}^{(l_1)} \otimes \mathbf{y}^{(l_2)}\right)_{m_3}^{l_3} = \sum_{m_1=-l_1}^{l_1} \sum_{m_2=-l_2}^{l_2} C_{m_3, m_2, m_1}^{l_3, l_2, l_1} x_{m_1}^{l_1} y_{m_2}^{l_2}, \tag{7}$$

where $C_{m_3, m_2, m_1}^{l_3, l_2, l_1}$ are Clebsch-Gordan coefficients (CGCs) [55]. The short-hand notation $\mathbf{x}^{(l_1)} \underset{l_3}{\otimes} \mathbf{y}^{(l_2)}$ is used to refer to the irrep of degree $l_3$ in the direct sum representation of $\mathbf{x}^{(l_1)} \otimes \mathbf{y}^{(l_2)}$. In other words,

the operation $\mathbf{x}^{(l_1)} \underset{l_3}{\otimes} \mathbf{y}^{(l_2)}$ performs the tensor product and contracts the result to a single irrep of degree $l_3$. A similar construction is also used in Clebsh-Gordan [56] and Tensor Field networks [42].

**Tensor product expansions** are inverse to tensor product contractions. Instead of contracting two irreps into one, CGCs are used to expand a single irrep $\mathbf{x}^{(l_3)}$ into a $(2l_1 + 1) \times (2l_2 + 1)$ matrix that represents its contribution to the direct sum representation of the tensor product of two irreps of degree $l_1$ and $l_2$, where $|l_2 - l_1| \le l_3 \le l_2 + l_1$:

$$\left( \overline{\otimes} \mathbf{x}^{(l_3)} \right)^{l_1, l_2}_{m_1, m_2} = \sum_{m_3 = -l_3}^{l_3} \mathbf{C}^{l_1, l_2, l_3}_{m_l, m_2, m_3} x^{l_3}_{m_3} . \tag{8}$$

Mirroring the shorthand used for tensor product contractions, $\overset{l_1, l_2}{\otimes} \mathbf{x}^{(l_3)}$ will be used to refer to the $(2l_1 + 1) \times (2l_2 + 1)$ matrix that is obtained from the tensor product expansion of $\mathbf{x}^{(l_3)}$.

**Selfmix layers** are used to recombine ("mix") the $F$ features of a single input $\mathbf{x} \in \mathbb{R}^{F \times (L_{\text{in}} + 1)^2}$ across different degrees and optionally allow changing the maximum degree from $L_{\text{in}}$ to $L_{\text{out}}$. The output features of degree $l_3$ are given by

$$\text{selfmix}_{L_{\text{in}} \to L_{\text{out}}}(\mathbf{x})^{(l_3)} = \begin{cases} \mathbf{k}_{l_3} \circ \mathbf{x}^{(l_3)} + \sum_{l_1=0}^{L_{\text{in}}} \sum_{l_2=l_1+1}^{L_{\text{in}}} \mathbf{s}_{l_3, l_2, l_1} \circ \left( \mathbf{x}^{(l_1)} \underset{l_3}{\otimes} \mathbf{x}^{(l_2)} \right) & l_3 \le L_{\text{in}} \\ \sum_{l_1=0}^{L_{\text{in}}} \sum_{l_2=l_1+1}^{L_{\text{in}}} \mathbf{s}_{l_3, l_2, l_1} \circ \left( \mathbf{x}^{(l_1)} \underset{l_3}{\otimes} \mathbf{x}^{(l_2)} \right) & l_3 > L_{\text{in}} \end{cases} . \tag{9}$$

Here, $\mathbf{k}, \mathbf{s} \in \mathbb{R}^F$ are learnable coefficients and the subscripts are used to distinguish independent parameters $\mathbf{k}, \mathbf{s}$ for different degrees: In total, a selfmix layer has $L_{\text{out}} + 1$ different $\mathbf{k}_{l_3}$ (one for each possible value of $l_3 \in \{0, \dots, L_{\text{out}}\}$) and $(L_{\text{out}} + 1) \frac{L_{\text{in}}(L_{\text{in}} + 1)}{2}$ different $\mathbf{s}_{l_3, l_2, l_1}$ (one for each valid combination of $l_3, l_2, l_1$).

**Spherical linear layers** Spherical linear layers are a combination of linear (Eq. 6) and selfmix (Eq. 9) layers given by

$$\text{sphlinear}_{L_{\text{in}} \to L_{\text{out}}, F_{\text{in}} \to F_{\text{out}}}(\mathbf{x}) = \text{linear}_{F_{\text{in}} \to F_{\text{out}}} \left( \text{selfmix}_{L_{\text{in}} \to L_{\text{out}}}(\mathbf{x}) \right) . \tag{10}$$

Chaining both operations allows arbitrary combinations across feature channels and degrees while still preserving rotational equivariance. In principle, whenever $L_{\text{in}} = L_{\text{out}}$, selfmix layers are not strictly necessary and Eq. 10 may be replaced by Eq. 6 for a boost in computational efficiency and reduction of memory footprint. However, this comes at the cost of reduced accuracy (see Section D in the supplement for details).

**Residual blocks** are modules consisting of two sequential spherical linear layers (see Eq. 10) and activation functions (see Eqs. 4 and 5) inspired by the pre-activation residual block described in [57]:

$$\text{residual}(\mathbf{x}) = \mathbf{x} + \text{sphlinear}_2(\boldsymbol{\sigma}_2(\text{sphlinear}_1(\boldsymbol{\sigma}_1(\mathbf{x})))) . \tag{11}$$

Here, $F_{\text{in}} = F_{\text{out}}$ and $L_{\text{in}} = L_{\text{out}}$ for both spherical linear layers.

**Pairmix layers** are used to combine a pair of features $\mathbf{x} \in \mathbb{R}^{F \times (L_x + 1)^2}$ and $\mathbf{y} \in \mathbb{R}^{F \times (L_y + 1)^2}$ with a scalar $r$ (e.g. their Euclidean distance) to generate new features of degree $L_{\text{out}}$:

$$\text{pairmix}_{L_x, L_y \to L_{\text{out}}}(\mathbf{x}, \mathbf{y}, r)^{(l_3)} = \sum_{l_1=0}^{L_x} \sum_{l_2=0}^{L_y} (\mathbf{W}_{l_3, l_2, l_1} \mathbf{g}(r)) \circ \left( \mathbf{x}^{(l_1)} \underset{l_3}{\otimes} \mathbf{y}^{(l_2)} \right) . \tag{12}$$

Here, $\mathbf{g}(r) \in \mathbb{R}^K$ is the vector $[g_0(r) \; g_1(r) \; \dots \; g_{K-1}(r)]^\top$ and $g_k(r)$ are radial basis functions. In this work, exponential Bernstein polynomials [27] are used (see Section B.2 and Eq. S9 in the supplement). The weight matrices $\mathbf{W} \in \mathbb{R}^{F \times K}$ allow to learn radial functions as linear combinations of the basis functions $g_k(r)$ and subscripts are used to distinguish independent weights for different combinations of $l_1, l_2, l_3$ (in total, there are $(L_x + 1)(L_y + 1)(L_{\text{out}} + 1)$ possible combinations).

**Interaction blocks** use message-passing to model interactions between the features $\mathbf{c} \in \mathbb{R}^{F \times (L+1)^2}$ of a central point $i$ with features $\mathbf{n} \in \mathbb{R}^{F \times (L+1)^2}$ of neighboring points $j$ within a local environment:

$$\mathbf{a}(\mathbf{x}, \mathbf{r})^{(l)} = \mathbf{x}^0 \circ (\mathbf{W}_l \mathbf{g}(\|\mathbf{r}\|)) \circ \mathrm{sphlinear}_{L \to L, 1 \to F}(\mathbf{Y}(\mathbf{r}))^{(l)},$$

$$\mathbf{b}(\mathbf{x}, \mathbf{r})^{(l)} = \mathrm{pairmix}_{L, L \to L}(\mathbf{x}, \mathrm{sphlinear}_{L \to L, 1 \to F}(\mathbf{Y}(\mathbf{r})), \|\mathbf{r}\|)^{(l)}, \tag{13}$$

$$\mathrm{interaction}(\mathbf{c}, \mathbf{n}, \mathbf{r})_i^{(l)} = \mathbf{c}_i^{(l)} + \sum_{j \neq i} \left( \mathbf{a}(\mathbf{n}_j, \mathbf{r}_{ij})^{(l)} + \mathbf{b}(\mathbf{n}_j, \mathbf{r}_{ij})^{(l)} \right).$$

Here $\mathbf{r}_{ij}$ is the distance vector $\mathbf{r}_{ij} = \mathbf{r}_j - \mathbf{r}_i$ between the positions $\mathbf{r}_i, \mathbf{r}_j$ of $i$ and $j$. The radial basis function expansion $\mathbf{g}$ is the same as in $\mathrm{pairmix}$ layers and $\mathbf{W} \in \mathbb{R}^{F \times K}$ are independent weight matrices for each degree $l$. Since geometric information enters Eq. 13 via relative distance vectors $\mathbf{r}_{ij}$ expanded in a spherical harmonics basis, interactions blocks are equivariant with respect to the SE(3) group of roto-translations.

## 4.2 PhiSNet architecture

PhiSNet takes as inputs nuclear charges $Z$ and positions $\mathbf{r}$ of $N$ atoms, which are used to construct equivariant feature representations encoding information about the chemical environment of each atom. These features are then further transformed and used to predict the entries of the Hamiltonian matrix, see below. An overview over the complete architecture is shown in Fig. 1B and more detailed diagrams of individual building blocks are given in Fig. S2 (see Section D of the supplement for ablation studies that explore the impact of possible simplifications of the PhiSNet architecture on prediction accuracy).

**Atomic feature representations** An embedding layer produces initial atomic feature representations $\mathbf{x}$ from the nuclear charges $Z$ according to

$$\mathrm{embedding}(Z)^{(l)} = \begin{cases} \mathbf{W} \mathbf{d}_Z + \mathbf{b}_Z & l = 0 \\ \mathbf{0} & l > 0 \end{cases}, \tag{14}$$

where $\mathbf{W} \in \mathbb{R}^{F \times 4}$ is a weight matrix and $\mathbf{b}_Z$ element-specific biases with learnable parameters. Here, $\mathbf{d}_Z \in \mathbb{R}^4$ are fixed vectors for each element that contain information about their nuclear charge and ground state electron configuration, similar to the embeddings described in [27] (see Section C.1 in the supplement for details). The features are then refined by five sequential modules, each consisting of identical building blocks with independent parameters:

$$\begin{aligned}
\mathbf{t} &= \mathrm{residual}(\mathbf{x}), \\
\mathbf{i} &= \mathrm{sphlinear}_{L \to L, F \to F}(\boldsymbol{\sigma}(\mathrm{residual}(\mathbf{t}))), \\
\mathbf{j} &= \mathrm{sphlinear}_{L \to L, F \to F}(\boldsymbol{\sigma}(\mathrm{residual}(\mathbf{t}))), \\
\mathbf{v} &= \mathrm{sphlinear}_{L \to L, F \to F}(\boldsymbol{\sigma}(\mathrm{residual}(\mathrm{interaction}(\mathbf{i}, \mathbf{j}, \mathbf{r})))), \\
\tilde{\mathbf{x}} &= \mathrm{residual}(\mathbf{t} + \mathbf{v}), \\
\tilde{\mathbf{y}} &= \mathrm{residual}(\tilde{\mathbf{x}}).
\end{aligned} \tag{15}$$

Each module produces two different outputs $\tilde{\mathbf{x}}$ and $\tilde{\mathbf{y}}$. The first output $\tilde{\mathbf{x}}$ serves as input to the next module in the chain (replacing $\mathbf{x}$ in Eq. 15), whereas the second output $\tilde{\mathbf{y}}$ is summed with the outputs of other modules $m$ to form the final atomic feature representations $\mathbf{f} = \sum_m \tilde{\mathbf{y}}_m$ (see Fig. S2A for a visual representation).

**Hamiltonian matrix prediction** The Hamiltonian matrix is constructed block-by-block, with each block corresponding to the interaction between two atoms $i$ and $j$. Diagonal and off-diagonal blocks are treated separately, i.e. different atomic pair features are constructed to predict them. Since these transformations also involve interactions with neighboring atoms (similar to interaction blocks), separate representations for central $\mathbf{c}$ and neighboring atoms $\mathbf{n}$ are created from the atomic features $\mathbf{f}$:

$$\begin{aligned}
\mathbf{c}_i &= \mathrm{residual}(\mathbf{f}_i), \\
\mathbf{n}_j &= \mathrm{residual}(\mathbf{f}_j).
\end{aligned} \tag{16}$$

Self-interaction features $\mathbf{f}_{ii}$ (for diagonal blocks) are constructed according to

$$\mathbf{f}_{ii}^{(l)} = \text{residual}\left(\mathbf{c}_i^{(l)} + \sum_{j\neq i}\left(\mathbf{n}_j^{(l)} \circ \mathbf{W}_l\mathbf{g}(\|\mathbf{r}_{ij}\|)\right)\right), \tag{17}$$

where $\mathbf{r}_{ij}$ is the distance vector between the positions of atoms $i$ and $j$, and the radial basis function expansion $\mathbf{g}$ is the same as in $\text{pairmix}$ layers. The matrices $\mathbf{W} \in \mathbb{R}^{F\times K}$ are separate trainable weight matrices for each degree $l$.

Similarly, pair-interaction features $\mathbf{f}_{ij}$ (for off-diagonal blocks) are obtained by combining the central representations of atoms $i$ and $j$ and interacting them with the neighbors of atom $i$ according to

$$\mathbf{f}_{ij}^{(l)} = \text{residual}\left(\text{pairmix}(\mathbf{c}_i, \mathbf{c}_j, \|\mathbf{r}_{ij}\|)^{(l)} + \sum_{k\notin\{i,j\}}\left(\mathbf{n}_k^{(l)} \circ \mathbf{W}_l\mathbf{g}(\|\mathbf{r}_{ik}\|)\right)\right). \tag{18}$$

All blocks $\mathbf{H}_{ii}$ and $\mathbf{H}_{ij}$ of the Hamiltonian matrix, each representing the interaction between two atoms, are themselves composed of smaller blocks corresponding to the interaction between atomic orbitals. Since atomic orbitals are expressed in a spherical harmonics basis, their interactions transform non-trivially (put predictably) under rotations. The correct equivariant behavior of a matrix block $\mathbf{M}^{l_1,l_2} \in \mathbb{R}^{(2l_1+1)\times(2l_2+1)}$ corresponding to the interaction between orbitals of degree $l_1$ and $l_2$ can be constructed as a sum over matrices obtained from tensor product expansions (Eq. 8) of irreps $\mathbf{a}$ (collected from specific channels of the pairwise features $\mathbf{f}_{ii}$ or $\mathbf{f}_{ij}$, see below) of all valid degrees $l_3 \in \{|l_2 - l_1|, \ldots, \leq l_2 + l_1\}$:

$$\mathbf{M}^{l_1,l_2} = \sum_{l_3=|l_2-l_1|}^{l_2+l_1} \overset{l_1,l_2}{\otimes} \mathbf{a}^{(l_3)}. \tag{19}$$

Two sets of indices $I^{\text{self}}$ and $I^{\text{pair}}$ count and keep track of the irreps necessary to construct all required matrices $\mathbf{M}^{l_1,l_2}$. For diagonal blocks $\mathbf{H}_{ii}$, the irreps of a given degree $l$ for the interaction of orbitals $n$ and $m$ of atoms with nuclear charge $Z$ are collected from specific channels of the self-interaction features $\mathbf{f}_{ii}$ via a unique index $I^{\text{self}}(Z, n, m, L)$. Similarly, for off-diagonal blocks $\mathbf{H}_{ij}$, a unique index $I^{\text{pair}}(Z_i, Z_j, n_i, n_j, L)$ selects irreps corresponding to orbitals $n_i$ and $n_j$ of atom pairs with nuclear charges $Z_i$ and $Z_j$ from the pair-interaction features $\mathbf{f}_{ij}$. After all blocks have been constructed, the complete matrix $\tilde{\mathbf{H}}$ is obtained by placing individual blocks at the appropriate positions (based on which atoms and orbitals interact). Finally, a Hamiltonian matrix satisfying the necessary Hermitian symmetry is constructed as $\mathbf{H} = \tilde{\mathbf{H}} + \tilde{\mathbf{H}}^T$. This symmetrization guarantees that both pair-features $\mathbf{f}_{ij}$ and $\mathbf{f}_{ji}$ contribute equally to the corresponding off-diagonal blocks and also makes sure that sub-blocks of the Hamiltonian swap positions in the correct way when equivalent atoms are permuted. In cases where multiple Hamiltonian-like matrices need to be predicted, all parameters up to the final residual blocks in Eqs. 17 and 18 are shared. An exception are overlap matrices (see Section 3), for which simpler self- and pair-interaction features derived directly from the embeddings are sufficient (see Section C.3 in the supplement for details). The complete block-wise construction process of the Hamiltonian matrix from irreps is illustrated in Fig. S5 for a water molecule with minimal basis set.

## 5 Results and discussion

To assess the ability of PhiSNet to predict molecular wavefunctions and electronic densities (see Fig. 1C), we train it on Kohn-Sham (the Kohn-Sham matrix takes the role of the Hamiltonian in DFT methods, see Section A.1 in the supplement) and overlap matrices for various non-equilibrium configurations of water, ethanol, malondialdehyde, uracil, and aspirin computed at the density functional theory (DFT) level with PBE/def2-SVP. Datasets for all molecules are taken from [33], with the exception of aspirin, for which geometries were sampled from the MD17 dataset [5] (more details on the datasets, training procedure, and hyperparameter settings can be found in Sections F and G of the supplement).

The results are summarized in Tab. 5 and compared to the current state-of-the-art given by SchNOrb [33]. PhiSNet achieves accuracy improvements up to two orders of magnitude, with

Table 1: Prediction errors of PhiSNet for various molecules compared to SchNOrb [33]. In addition to Kohn-Sham $\mathbf{K}$ and overlap $\mathbf{S}$ matrices, we also report errors for energies $\epsilon$ and the cosine similarity between predicted and reference wavefunction $\psi$ for all occupied orbitals. Best results in bold.

| Data set | | $\mathbf{K}$ [$10^{-6}$ E$_\text{h}$] | $\mathbf{S}$ [$10^{-6}$] | $\epsilon$ [$10^{-6}$ E$_\text{h}$] | $\psi$ |
|---|---|---|---|---|---|
| Water | SchNOrb | 165.4 | 79.1 | 279.3 | **1.00** |
| | PhiSNet | **17.59** | **1.56** | **85.53** | **1.00** |
| Ethanol | SchNOrb | 187.4 | 67.8 | 334.4 | **1.00** |
| | PhiSNet | **12.15** | **0.626** | **62.75** | **1.00** |
| Malondialdehyde | SchNOrb | 191.1 | 67.3 | 400.6 | 0.99 |
| | PhiSNet | **12.32** | **0.567** | **73.50** | **1.00** |
| Uracil | SchNOrb | 227.8 | 82.4 | 1760 | 0.90 |
| | PhiSNet | **10.73** | **0.533** | **84.03** | **1.00** |
| Aspirin | SchNOrb | 506.0 | 110 | 48689 | 0.57 |
| | PhiSNet | **12.84** | **0.406** | **176.6** | **0.98** |

the biggest differences arising in larger and more complex molecules like uracil and aspirin. Note that the training process for SchNOrb requires data augmentation via random rotations to approximate the equivariance relation between wavefunction and molecular orientation, while PhiSNet preserves exact equivariance. This not only allows for faster convergence, but also leads to much smaller model sizes, with our model requiring approximately one fifth of the parameters of SchNOrb while providing significantly more accurate results. In addition, PhiSNet provides speedups of over three orders of magnitude compared to DFT calculations (see Section E.1 for details).

The improved prediction accuracy provided by PhiSNet makes it possible to accurately derive properties such as energies and forces directly from the wavefunction in an end-to-end manner, enabling a number of interesting and novel ML applications for the molecular sciences. As an example, we showcase a transfer-learning application, where a model trained on low accuracy Hartree-Fock (HF) electronic structure calculations is fine-tuned to learn a correction to the wavefunction, such that energies and forces match those obtained via high-level coupled cluster with singles, doubles, and perturbative triple excitations (CCSD(T)) calculations. The CCSD(T) method models the effects of electronic many-body interactions, which are neglected in HF theory, and is often considered to be the "gold standard" of quantum chemistry [8]. However, its accuracy comes at a significantly increased computational complexity, going from $\mathcal{O}(N^3)$ for HF to $\mathcal{O}(N^7)$ for CCSD(T) (here, $N$ is the number of basis functions). Thus, a machine-learned correction to HF theory, which mimics the effects of electron correlation in a computationally efficient manner, is a possible way towards novel hybrid methods that rival the accuracy of high level electronic structure calculations and combine the generality and robustness of *ab initio* methods with the efficiency of ML.

After pretraining PhiSNet on HF/cc-pVDZ data for Fock matrices $\mathbf{F}$, core Hamiltonians $\mathbf{H}^{\text{core}}$, and overlap matrices $\mathbf{S}$, we fine-tune it on forces computed at the CCSD(T)/cc-pVTZ level, but still keep loss terms for $\mathbf{H}^{\text{core}}$ and $\mathbf{S}$ computed with HF/cc-pVDZ (see Section A.1 in the supplement for a brief overview of HF theory, where we explain the relevance of $\mathbf{F}$, $\mathbf{H}^{\text{core}}$, and $\mathbf{S}$, and how energies and forces are derived from them). This way, the model learns to adapt only the Fock matrix $\mathbf{F}$, which embodies the electron-electron interactions in the HF formalism. Interestingly, very subtle changes to $\mathbf{F}$ seem to be sufficient to approximate the effects of electron correlation, resulting in an "effective CCSD(T) wavefunction" that, at first glance, appears to be almost identical to its original HF-level counterpart (see Fig. 1D). Nonetheless, the modified wavefunction reduces the mean absolute errors (MAEs) between energies and forces predicted with PhiSNet to just 79 $\mu$E$_\text{h}$ and 0.85 mE$_\text{h}$a$_0^{-1}$, respectively, compared to the CCSD(T) reference. In contrast, the original HF-level wavefunction leads to MAEs of 4266 $\mu$E$_\text{h}$ and 15 mE$_\text{h}$a$_0^{-1}$ for energies and forces, respectively, i.e. prediction errors are reduced by over one order of magnitude with no additional computational overhead. More details on the transfer learning application can be found in Section G.2 of the supplement.

Although PhiSNet predicts electronic properties very accurately, some applications might require *exact* solutions, making existing quantum chemistry methods preferable over predictions of an

ML model. Even then, PhiSNet can be used to achieve speedups without any loss of accuracy or robustness: A significant fraction of compute time in HF and DFT calculations must be spent to arrive at a self-consistent solution of Eq. 3. A good initial guess for the wavefunction can reduce the number of required iterations significantly. We observe a decrease between 56–72% in the number of iterations, resulting in a total reduction of wall-clock time between 24–40%, when using PhiSNet-predicted wavefunctions in place of the default guess (see Section E.2 in the supplement for more details).

While the results are highly promising, the current approach also has limitations. Due to the fact that all pairwise combinations of atoms have to be considered for constructing the pair-interaction features $\mathbf{f}_{ij}$, the Hamiltonian matrix prediction scales quadratically with the number of atoms. Further, to derive the coefficient matrix $\mathbf{C}$ defining the wavefunction, a generalized eigenvalue problem has to be solved (see Eq. 3), which scales cubically with the matrix size (the number of basis functions). For these reasons, PhiSNet does not scale well to systems with a very large number of atoms. However, possible extensions of our method could exploit the fact that orbital overlaps between distant atoms are very small, so the entries of corresponding matrix blocks are approximately zero [58]. Then, only pair-interaction features for non-zero blocks need to be computed and the solution of Eq. 3 can exploit sparsity.

# 6 Conclusion

For learning problems with known invariances, equivariances, symmetries, or other constraints, as is common in physics applications, it is useful to include such properties directly into the model architecture. This effectively reduces the complexity of the learning problem (cf. [59, 60]), increasing model performance and decreasing the required amount of training data. While invariance and equivariance properties could also be (approximately) learned, this typically requires much more reference data and/or data augmentation. In contrast, by "hard coding" domain knowledge, the learning is constrained to a meaningful submanifold, e.g. reflecting rotational equivariance [38], energy conservation, physical laws or symmetries (e.g. [4, 5, 23, 27, 61]), group equivariance (e.g. [37]), graph properties (e.g. [40, 46]) or alike. Thus, known properties do not need to be learned explicitly, because the submanifold where learning takes place already embodies them.

Our present contribution follows this design principle, specifically, we describe a series of general SE(3)-equivariant operations and building blocks for deep learning architectures operating on geometric point cloud data, which we used here to construct PhiSNet, a novel neural network architecture capable of accurately predicting wavefunctions and electronic densities. Unlike previous models, which need to approximately learn how the wavefunction transforms under molecular rotations and rely on data augmentation, the SE(3)-equivariant[3] building blocks of our network allow to exactly capture the correct transformation without needing to learn it explicitly. By applying PhiSNet on a range of small to medium-sized molecules, we demonstrated that our model achieves accuracy improvements of up to two orders of magnitude compared to the previous state-of-the-art (while at the same time requiring significantly less parameters), and speedups with respect to *ab initio* solutions of over three orders of magnitude.

For the first time, sufficient accuracy is reached to derive quantum mechanical observations directly from the predicted wavefunction in an end-to-end manner, which also allows to adapt the predicted wavefunction such that it leads to desired physical properties. To showcase such an application, we fine-tuned a model trained on low accuracy wavefunctions to predict properties computed at a much higher level of theory, thereby learning a correction that implicitly captures the effects of many-body electron correlation. This paves the way towards the development of novel semi-empirical methods that are capable of providing highly accurate quantum chemical calculations at a drastically reduced computational cost. In addition, we demonstrated that existing *ab initio* approaches can benefit from guess wavefunctions provided by our method, achieving significant speedups without any loss of accuracy or robustness.

Although we focus on quantum chemistry applications in this contribution, we would like to reiterate that the presented SE(3)-equivariant operations are general and can be used to construct other deep learning architectures for geometric point cloud data beyond physics, e.g. [37, 38, 40, 45, 46].

---

[3]Note that SE(3)-equivariance is important (in contrast to just SO(3)-equivariance), because the properties of chemical systems may change substantially when individual atoms are translated relative to each other.

## Acknowledgments and Disclosure of Funding

OTU acknowledges funding from the Swiss National Science Foundation (Grant No. P2BSP2_188147). MB acknowledges support by the Federal Ministry of Education and Research (BMBF) for BIFOLD (01IS18037A). KRM was supported in part by the Institute of Information & Communications Technology Planning & Evaluation (IITP) grant funded by the Korea Government (No. 2019-0-00079, Artificial Intelligence Graduate School Program, Korea University), and was partly supported by the German Ministry for Education and Research (BMBF) under Grants 01IS14013A-E, 01GQ1115, 01GQ0850, 01IS18025A and 01IS18037A; the German Research Foundation (DFG) under Grant Math+, EXC 2046/1, Project ID 390685689. M. Gastegger works at the BASLEARN Joint Lab for Machine Learning, co-financed by TU Berlin and BASF SE. There are no competing interests to declare.

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
