# Supplementary information for SE(3)-equivariant prediction of molecular wavefunctions and electronic densities

## A   Detailed Background

### A.1   Quantum chemistry

At the center of quantum chemistry methods lies the electronic Schrödinger equation

$$\hat{H}_{\text{el}}\Psi_{\text{el}} = E_{\text{el}}\Psi_{\text{el}}\,, \tag{S1}$$

which describes the physical laws underlying the interactions between nuclei and electrons. Here, $\hat{H}_{\text{el}}$ is the electronic Hamiltonian operator which describes effects due to the kinetic energy of the electrons, the interactions between electrons and nuclei, as well as the inter-electronic interactions. The wavefunction $\Psi_{\text{el}}$, an eigenfunction of $\hat{H}_{\text{el}}$, captures the spatial distribution of electrons and the corresponding eigenvalue $E_{\text{el}}$ represents the electronic energy of the system.

Before the eigenvalue problem can be solved, a suitable functional expression for $\Psi_{\text{el}}$ has to be found. A standard approach is to express the wavefunction as a Slater determinant $\Psi_{\text{el}} = |\psi_1 \ldots \psi_n\rangle$, an anti-symmetric product of molecular orbitals $\psi_i$, which are constructed as linear combinations of atom-centered basis functions $\psi_i = \sum_j C_{ij}\phi_j$. These atomic orbitals $\phi_j$ are typically taken to be products of radial functions $R_l$ and spherical harmonics $Y_l^m$

$$\phi_i(\mathbf{r}) = R_l(|\mathbf{r}|)Y_l^m(\mathbf{r}), \tag{S2}$$

where $\mathbf{r}$ denotes the electronic coordinates. Using this ansatz for the wavefunction leads to

$$\mathbf{H}\mathbf{C} = \epsilon\mathbf{S}\mathbf{C}\,, \tag{S3}$$

where the Hamiltonian is written as a matrix $\mathbf{H}$ with entries $H_{ij} = \int \phi_i^*(\mathbf{r})\hat{H}_{\text{el}}\phi_j(\mathbf{r})d\mathbf{r}$. The overlap matrix $\mathbf{S}$ with entries $S_{ij} = \int \phi_i^*(\mathbf{r})\phi_j(\mathbf{r})d\mathbf{r}$ has to be introduced and a generalized eigenvalue problem must be solved, because the basis functions $\phi$ are usually not orthonormal. The eigenvectors $\mathbf{C}$ specify the wavefunction $\Psi_{\text{el}}$ via the coefficients $C_{ij}$ of the molecular orbitals $\psi_i$ and the eigenvalues $\epsilon$ are the corresponding orbital energies. Unfortunately, the entries of $\mathbf{H}$ depend on $\mathbf{C}$, because the many-body inter-electronic interactions depend on the positions of all electrons. In other words, $\mathbf{H}$ cannot be determined without knowing $\mathbf{C}$, which in turn cannot be determined without knowing $\mathbf{H}$. To still be able to solve Eq. S3, approximations have to be introduced.

In the Hartree-Fock formalism, also known as self-consistent field (SCF) method, the Hamiltonian is replaced by the Fock matrix $\mathbf{F} = \mathbf{H}^{\text{core}} + \mathbf{G}$, which consists of a one-electron core Hamiltonian $\mathbf{H}^{\text{core}}$ (describing the kinetic energy of the electrons and their interaction with nuclei) and a two-electron part $\mathbf{G}$. Then, starting from an initial guess for $\mathbf{C}$, the matrix $\mathbf{G}$ is computed by approximating the true electron-electron interaction by letting each electron interact with the mean field caused by all other electrons (neglecting correlation effects). The resulting Fock matrix $\mathbf{F}$ is used to solve Eq. S3 (replacing $\mathbf{H}$), leading to updated coefficients $\mathbf{C}$. The two-electron part $\mathbf{G}$ of the Fock matrix is updated using the newly determined coefficients and the procedure is repeated until a self-consistent solution is found. Once a converged solution is found, the total ground state energy $E$ of the chemical system is obtained as

$$E = \sum_{i \in \phi^{\text{occ}}} (\epsilon_i^{\text{core}} + \epsilon_i) + \frac{1}{2}\sum_{I,J} \frac{Z_I Z_J}{\|\mathbf{R}_{IJ}\|}. \tag{S4}$$

The first sum runs over all occupied (lowest-energy) orbitals, where $\epsilon_i$ are the entries of $\boldsymbol{\epsilon}$ (see Eq. S3) and $\epsilon_i^{\mathrm{core}} = \mathrm{diag}(\mathbf{C}^*\mathbf{H}^{\mathrm{core}}\mathbf{C})$. The second term accounts for the classical Coulomb repulsion between nuclei, where $Z_I, Z_J$ are the nuclear charges and $\|\mathbf{R}_{IJ}\|$ is the distance between two nuclei. The forces acting on nuclei can be computed by differentiating the energy in Eq. S4 with respect to the nuclear coordinates, i.e. the force acting on nucleus $I$ is given by $-\frac{\partial E}{\partial \mathbf{R}_I}$.

The computational cost of the HF method scales $\mathcal{O}(N^3)$ with the number of basis functions $N$ and is dominated by the iterative procedure and the need to re-evaluate the matrix $\mathbf{F}$ (more precisely, the two-electron part $\mathbf{G}$) whenever the coefficients $\mathbf{C}$ change, which is costly since it involves two-electron integrals. The biggest downside of the HF solution is its low accuracy due to the neglect of electron correlation. To overcome this limitation, so-called post-HF methods like coupled cluster theory have been developed, using the HF wavefunction as a starting point. However, the improved accuracy of such approaches comes at a significantly higher computational cost (for example $\mathcal{O}(N^7)$), making them prohibitively expensive for large chemical systems.

An efficient alternative to HF and post-HF methods is density functional theory (DFT), where the wavefunction is replaced by the electron density. In this framework, electron correlation can in theory be treated exactly via the so-called exchange-correlation functional. While DFT scales $\mathcal{O}(N^3)$, the exact form of this functional is unknown and must be approximated with empirical functions, limiting the accuracy of DFT. Conveniently, DFT in its most frequently used formulation (Kohn-Sham DFT) can be cast in a similar matrix form as Eq. S3, where the Fock matrix is replaced by the Kohn-Sham matrix $\mathbf{K}$.

## A.2 Group representations and equivariance

A representation $D$ of a group $G$ is a function from $G$ to square matrices such that for all $g, h \in G$

$$D(g)D(h) = D(gh). \tag{S5}$$

A function $f : \mathcal{X} \mapsto \mathcal{Y}$, where $\mathcal{X}$ and $\mathcal{Y}$ are vector spaces, is called equivariant with respect to a group $G$ and representations $D^{\mathcal{X}}$ and $D^{\mathcal{Y}}$ if for all $g \in G, \mathbf{x} \in \mathcal{X}$, and $\mathbf{y} \in \mathcal{Y}$:

$$f(D^{\mathcal{X}}(g)\mathbf{x}) = D^{\mathcal{Y}}(g)f(\mathbf{x}). \tag{S6}$$

When $D^{\mathcal{Y}}(g)$ is the identity function, the function $f$ is said to be invariant with respect to $G$. Since the composition of two equivariant functions $f_1$ and $f_2$ is also equivariant, any deep neural network that is composed of layers of equivariant functions is itself equivariant.

In the case of the group of 3D rotations, known as SO(3), any representation $g \in \mathrm{SO}(3)$ can be decomposed as a direct sum of irreducible representations of dimension $2l + 1$, where we call $l$ the degree of the representation. The irreducible linear operators of SO(3) are known as Wigner-D matrices, with a Wigner-D matrix of degree $l$ being denoted as $\mathbf{D}^{(l)} \in \mathbb{R}^{(2l+1)\times(2l+1)}$.

## A.3 Spherical harmonics

Spherical harmonics $Y_l^m$ of degree $l = 0, \ldots, \infty$ and order $m = -l, \ldots, l$ form a complete orthonormal basis for functions on the surface of a sphere and are the irreducible representations (irreps) of the 3D rotation group SO(3). Let $\mathcal{R}(g) \in \mathbb{R}^{3\times3}$ represent the 3D rotation matrix corresponding to some group element $g \in \mathrm{SO}(3)$ and $\mathbf{D}^{(l)}(g) \in \mathbb{R}^{(2l+1)\times(2l+1)}$ denote the Wigner-D matrix of degree $l$ representing $g$, then for all $g \in \mathrm{SO}(3), \mathbf{r} \in \mathbb{R}^3$:

$$Y_l^m(\mathcal{R}(g)\mathbf{r}) = \sum_{m'} \mathbf{D}_{mm'}^{(l)}(g)Y_l^{m'}(\mathbf{r}). \tag{S7}$$

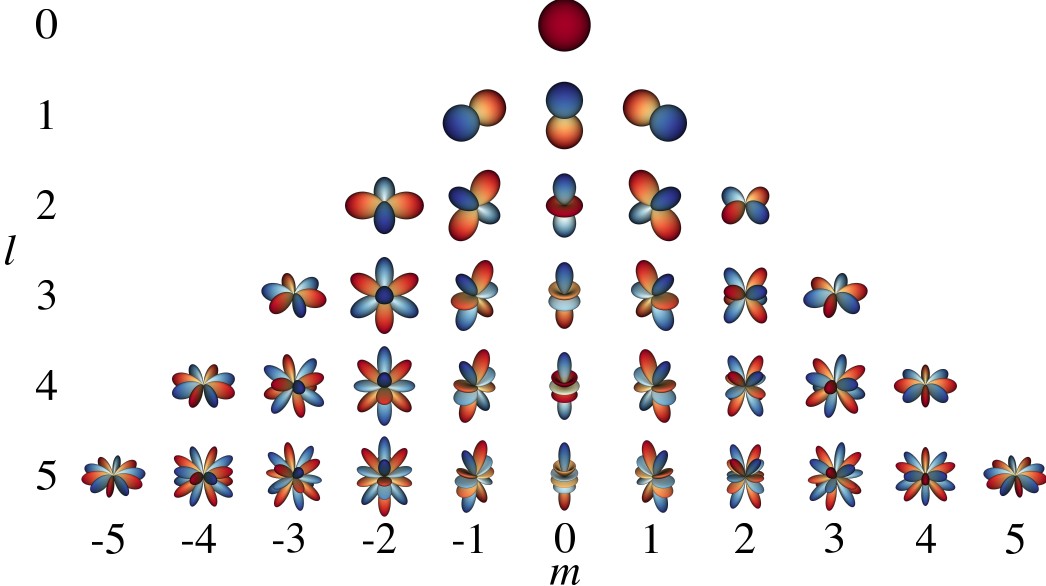

Figure S1: Visualization of spherical harmonics $Y_l^m$ (Eq. S8) of degree $l \in \{0, \ldots, 5\}$ and order $m \in \{-l, \ldots, l\}$ (red and blue indicate positive and negative values, respectively). There are $(L+1)^2$ different possible combinations of $l$ and $m$ for a maximum degree $L$.

In our implementation we use real-valued spherical harmonics, which for a given degree $l \geq 0$ and order $-l \leq m \leq l$ are defined as

$$
Y_l^m(\mathbf{r}) = \sqrt{\frac{2l+1}{2\pi}} \Pi_l^{|m|}(z)
\begin{cases}
\sum_{p=0}^{|m|} \binom{|m|}{p} x^p y^{|m|-p} \sin\left((|m|-p)\frac{\pi}{2}\right) & m < 0 \\
\frac{1}{\sqrt{2}} & m = 0 \\
\sum_{p=0}^{m} \binom{m}{p} x^p y^{m-p} \cos\left((m-p)\frac{\pi}{2}\right) & m > 0
\end{cases}
\tag{S8}
$$

$$
\Pi_l^m(z) = \sqrt{\frac{(l-m)!}{(l+m)!}} \sum_{k=0}^{\lfloor (l-m)/2 \rfloor} (-1)^k 2^{-l} \binom{l}{k} \binom{2l-2k}{l} \frac{(l-2k)!}{(l-2k-m)!} r^{2k-l} z^{l-2k-m},
$$

where $x$, $y$, and $z$ are the Cartesian components of vector $\mathbf{r} \in \mathbb{R}^3$ and $r = \|\mathbf{r}\|$.

## A.4  Group equivariance in machine learning

One of the first works to adopt a group theoretic approach to constructing equivariant neural networks was by Cohen and Welling [1], where equivariance for discrete finite groups is achieved by transforming the convolutional kernel or feature representation according to each group element and aggregating the results. While the method was initially developed for images, it could be extended to other types of data such as vector fields [2]. However, the method was limited to applications with discrete groups and a small number of group elements, since the kernels/features need to be explicitly transformed for every group element.

Cohen and Welling [3] also explored the possibility of representing convolution kernels as a linear combination of equivariant basis functions, introducing the general concept and demonstrating it for the discrete group of $90°$ rotations. Worrall et al. [4] used circular harmonics as a basis for the convolutional filters to achieve equivariance to the continuous SO(2) group of 2D rotations. This principle was soon extended to the SO(3) group of 3D rotations by using sphercal harmonics as basis functions [5–9], which allows applications to 3D structures such as spherical images, voxel data, and point clouds, including atomistic systems. Similarly, Kondor et al. [10] and Hy et al. [11] used

group theoretical principles to develop a permutationally equivariant graph neural network to predict properties for atomistic systems.

Most recently, there has been further theoretical research on equivariant neural networks, with Cohen et al. [12] extending the principle of equivariance from global symmetries to local gauge transformations, allowing the implementation of a very efficient alternative to spherical CNNs [6], Keriven and Peyré [13] providing a proof of the universailty of equivariant graph networks with a single hidden layer, and works such as [14, 15] aiming to provide a general framework for the analysis and construction of equivariant networks for a wide range of problems. Very recently, Fuchs et al. [16] introduced the SE(3)-transformer [16], an equivariant generalization of the popular transformer architecture [17].

In this work, we focus on introducing equivariant building blocks mainly used in the context of message-passing neural networks (MPNNs) [18], which are applicable to a variety of graph based problems. From this perspective, our work is most closely related to Cormorant [19] and tensor field networks and their variants [7, 20], which are rotationally equivariant MPNNs using spherical harmonics and SO(3) irreps as equivariant representations. However, the architectures, feature representations, and operations used in these works have some significant differences to some of the most successful rotationally invariant MPNNs for chemical applications (e.g. [21, 22]). When designing our proposed SE(3)-equivariant operations, the aim was to mimic the main building blocks of rotationally invariant MPNNs as closely as possible, such that existing architectures can be "translated" to an equivariant framework. Our hope was to benefit from design principles that proved to be successful in previous works, without the need to re-design a successful architecture around the concept of rotational equivariance from scratch. As a result, our proposed PhiSNet architecture (based on the rotationally invariant PhysNet [22]) has several key differences compared to previous equivariant MPNNs.

For example, Cormorant does not use explicit activation functions and instead relies on an operation similar to our proposed tensor product contractions as a source of non-linearities, which can cause difficulties during training. While tensor field networks apply activation functions to scalar features (similar to our work), couplings between features of different degrees are only possible in convolution layers, where two (or more) different representations are interacting. In contrast, the operations proposed in this work allow activation functions to affect all degrees of all feature channels through the use of $\mathrm{selfmix}$ layers. Our proposed spherical linear layer (which contains a $\mathrm{selfmix}$ operation) acts as a drop-in replacement of linear layers in invariant architectures and allows couplings between all channels and degrees of feature representations. Additionally, we introduce tensor product expansions of irreducible representations as a general way to construct highly complex second order tensors, while preserving their equivariance properties.

## B  Additional details on SE(3)-equivariant neural network building blocks

### B.1  Properties of the generalized SiLU activation function

Depending on the values of $\alpha$ and $\beta$, the generalized SiLU activation (Eq. 5) smoothly interpolates between a linear function and the popular ReLU activation [23] (see Fig. S3). The initial values are chosen as $\alpha = 1.0$ and $\beta = 1.702$ such that Eq. 5 approximates the GELU function [24].

### B.2  Exponential Bernstein polynomial basis functions

All distance-dependant SE(3)-equivariant operations (e.g. $\mathrm{pairmix}$ layers) rely on a vector $\mathbf{g} = [g_0(r)\, g_1(r)\, \ldots\, g_{K-1}(r)]^\top$ representing a basis expansion of the distance via exponential Bernstein radial basis functions [25] given by

$$
\begin{aligned}
g_k(r) &= b_{K-1,k}\left(e^{-\gamma r}\right) f_{\mathrm{cut}}(r) \\
b_{\nu,n}(x) &= \binom{n}{\nu} x^\nu (1-x)^{n-\nu} \qquad \nu = 0, \ldots, n \\
f_{\mathrm{cut}}(r) &= \begin{cases} \exp\left(-\dfrac{r^2}{(r_{\mathrm{cut}} - r)(r_{\mathrm{cut}} + r)}\right) & r < r_{\mathrm{cut}} \\ 0 & r \geq r_{\mathrm{cut}} \end{cases},
\end{aligned}
\tag{S9}
$$

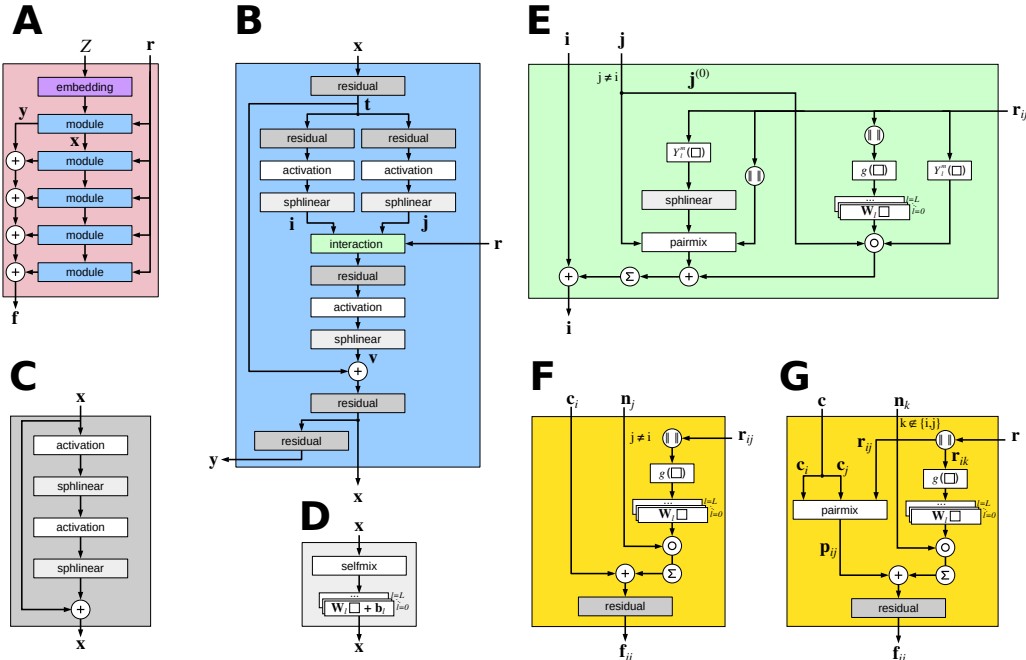

Figure S2: Illustration of PhiSNet components. **A**: Generation of atomic spherical harmonics features $\mathbf{f}$. The Cartesian coordinates $\{\mathbf{r}_i\}$ of the atoms are used to calculate a spherical harmonics representation of the relative atomic positions. An embedding layer (purple) creates initial atomic features from the nuclear charges $\{Z_i\}$, which are refined through a series of equivariant modular blocks (blue). The outputs $\mathbf{y}$ of each modular block are summed to obtain the final representations $\mathbf{f}$. **B**: Modular block. Input representations are separated into two branches, which produce separate features for the central $\mathbf{i}$ and neighboring atoms $\mathbf{j}$. These are coupled by an interaction block (green) and added to the original features to produce the $\mathbf{x}$ and $\mathbf{y}$ outputs. **C**: Residual blocks (Eq. 11) pass input features through two non-linear activations and spherical linear layers and add the result to the unmodified input via a skip-connection. **D**: Spherical linear layers (Eq. 10) are composed of a selfmix layer (Eq. 9) followed by separate linear layers for each spherical harmonic order and are used to re-combine spherical harmonic orders and feature channels. **E**: The interaction block (Eq. 13) encodes information about chemical environments by combining features of neighboring atoms with a spherical harmonics based representation of their relative position to a central atom. **F**: Self-interaction features are generated by updating central atom features with features of neighboring atoms, similar to the interaction block (Eq. 17). **G**: Pair-interaction features are obtained by combining the features of a pair of atoms with a pairmix layer and interacting them with the neighboring atoms of the first atom in each pair (Eq. 18).

where $b_{\nu,n}(x)$ are ordinary Bernstein basis polynomials. For $n \to \infty$, linear combinations of $b_{\nu,n}(x)$ approximate any continuous function on the interval $[0, 1]$ uniformly [26]. The transformation $x = e^{-\gamma r}$ maps distances $r$ from $[0, \infty]$ to $[0, 1]$ and introduces a chemically meaningful inductive bias, i.e. wave functions of electrons are known to decay exponentially with increasing distance from a nucleus (a similar mapping is also used in [22, 27]). The parameter $\gamma$ is a learnable "length scale" and the cutoff function $f_{\mathrm{cut}}(r)$ ensures that $g_k(r)$ smoothly goes to zero for $r \geq r_{\mathrm{cut}}$ (see Fig. S4). For computational efficiency, all radial basis function expansions across all layers share the same $\gamma$. This way, only one vector $\mathbf{g}(r)$ needs to be computed for each relevant distance $r$.

## C   Additional details on the PhiSNet architecture

### C.1   Element descriptors used in embeddings

The embeddings used in the PhiSNet architecture rely on element descriptors $\mathbf{d}_Z$, which encode information about the nuclear charge and the ground state configuration of each element. For example, oxygen ($Z = 8$) has the ground state configuration $1s^2 2s^2 2p^4$ and its corresponding descriptor is $\mathbf{v}_8 = [8\ 2\ 2\ 4]^\top$ (see Table S1 for more examples). In this work, only elements from periods 1 and 2 of the periodic table are considered, i.e. 4-dimensional descriptors (with entries for $Z$, 1s, 2s, and 2p electrons) are sufficient. To include elements from higher periods, the descriptors could be extended

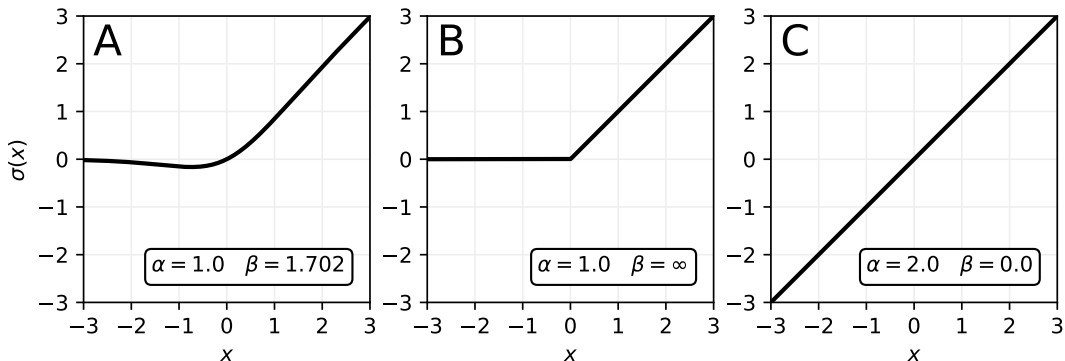

Figure S3: Generalized SiLU activation (see Eq. 5). (A) When the parameters are set to $\alpha = 1.0$, $\beta = 1.702$, Eq. 5 approximates the GELU function [24]. (B) For $\beta \to \infty$, $\sigma(x) \to \alpha \cdot \max(0, x)$, i.e. $\sigma(x)$ approaches a (scaled) ReLU activation [23]. (C) When $\beta$ is zero, Eq. 5 is a linear function with slope $\frac{\alpha}{2}$.

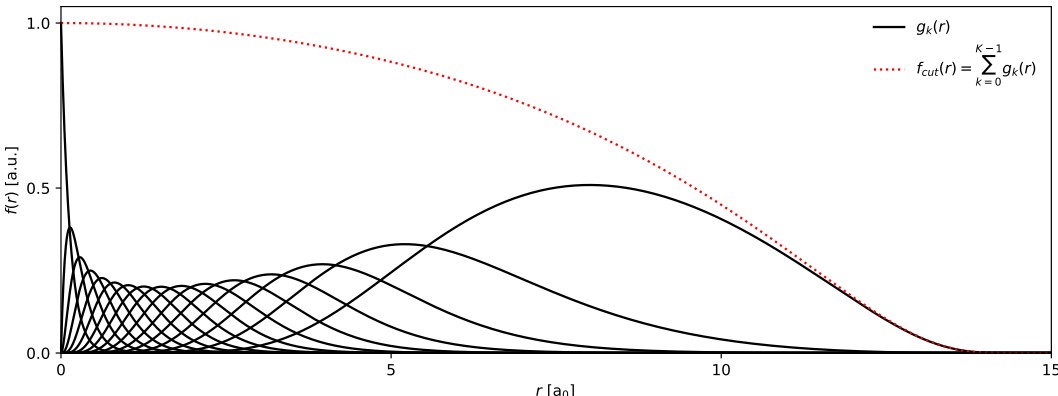

Figure S4: Exponential Bernstein basis functions $g_k(r)$ for $K = 16$, $\gamma = 0.5 \ \mathrm{a}_0^{-1}$, and $r_{\mathrm{cut}} = 15 \ \mathrm{a}_0$ (see Eq. S9). Since the Bernstein basis polynomials form a partition of unity, the sum $\sum_k g_k(r)$ is equal to $f_{\mathrm{cut}}(r)$.

with additional entries (for example for 3s and 3p electrons for period 3 elements). While the bias term $\mathbf{b}_Z$ in Eq. 14 offers sufficient degrees of freedom to learn arbitrary embeddings for different elements, including the term $\mathbf{W}\mathbf{d}_Z$ provides an inductive bias that takes into account the quantum chemical structure of different elements. Similar element descriptors are also used in [25].

| element | Z | 1s | 2s | 2p | $\mathbf{d}_Z^\top$ |
|---------|---|----|----|----|---------------------|
| H | 1 | 1 | 0 | 0 | $[1\ 1\ 0\ 0]$ |
| C | 6 | 2 | 2 | 2 | $[6\ 2\ 2\ 2]$ |
| N | 7 | 2 | 2 | 3 | $[7\ 2\ 2\ 3]$ |
| O | 8 | 2 | 2 | 4 | $[8\ 2\ 2\ 4]$ |

Table S1: Examples of element descriptors $\mathbf{d}_Z$.

## C.2 Details on the block-wise Hamiltonian matrix prediction

To illustrate the process of the block-wise Hamiltonian matrix construction for a concrete example, let us consider a water molecule using a minimal basis set, i.e. a single basis function is used to represent each atomic orbital. Water consists of two H atoms ($Z = 1$) with a 1s ($l = 0$) orbital, and one O atom ($Z = 8$) with a 1s ($l = 0$), a 2s ($l = 0$), and a 2p ($l = 1$) orbital. An orbital with degree $l$ has $2l + 1$ distinct orders $m$, so the complete Hamiltonian is a $7 \times 7$ matrix. In general, to represent the $(2l_1 + 1) \times (2l_2 + 1)$ matrix block that corresponds to the interaction between two orbitals of

degrees $l_1$ and $l_2$, we need irreps for all degrees $l_3 \in \{|l_1 - l_2|, ..., l_1 + l_2\}$. In the case of water with a minimal basis set, there are three main cases to consider:

1. Interaction between two s-orbitals ($l_1 = 0, l_2 = 0$), i.e. $\mathbb{1} \otimes \mathbb{1} = \mathbb{1}$ ($1 \times 1$ matrix block, an irrep of degree $l_3 = 0$ is needed)

2. Interaction between an s- and a p-orbital ($l_1 = 0, l_2 = 1$), i.e. $\mathbb{1} \otimes \mathbb{3} = \mathbb{3}$ ($1 \times 3$ matrix block, an irrep of degree $l_3 = 1$ is needed)

3. Interaction between two p-orbitals ($l_1 = 1, l_2 = 1$), i.e. $\mathbb{3} \otimes \mathbb{3} = \mathbb{1} \oplus \mathbb{3} \oplus \mathbb{5}$ ($3 \times 3$ matrix block, irreps of degrees $l_3 = 0, 1, 2$ are needed)

Now, unique channel indices have to be assigned to collect irreps of all degrees for the interaction between individual orbitals from the self-interaction and pair-interaction features $\mathbf{f}_{ii}$ and $\mathbf{f}_{ij}$. The indices are stored in the corresponding index sets $I^{\text{self}}$ and $I^{\text{pair}}$ and the total number of feature channels needs to be chosen large enough (such that all assigned indices are valid). To construct the diagonal blocks, 6 irreps of degree 0, 5 irreps of degree 1, and 1 irrep of degree 2 are necessary, whereas for the off-diagonal blocks, 5 irreps of degree 0, and 2 irreps of degree 1 are required (see Table S2 for a complete breakdown). Fig. S5 illustrates how the irreps are transformed to matrix blocks via tensor product expansions (Eq. 8) and accumulated to construct the complete Hamiltonian matrix.

| **irreps for diagonal blocks** | | | |
| $I^{\text{self}}(Z, n, m, L)$ | orbital interaction | degree | index |
|---|---|---|---|
| $I^{\text{self}}(8, 1, 1, 0)$ | O:1s $\times$ O:1s $\rightarrow \mathbb{1}$ | 0 | 1 |
| $I^{\text{self}}(8, 1, 2, 0)$ | O:1s $\times$ O:2s $\rightarrow \mathbb{1}$ | 0 | 2 |
| $I^{\text{self}}(8, 2, 1, 0)$ | O:2s $\times$ O:1s $\rightarrow \mathbb{1}$ | 0 | 3 |
| $I^{\text{self}}(8, 2, 2, 0)$ | O:2s $\times$ O:2s $\rightarrow \mathbb{1}$ | 0 | 4 |
| $I^{\text{self}}(8, 3, 3, 0)$ | O:2p $\times$ O:2p $\rightarrow \mathbb{1}$ | 0 | 5 |
| $I^{\text{self}}(1, 1, 1, 0)$ | H:1s $\times$ H:1s $\rightarrow \mathbb{1}$ | 0 | 6 |
| $I^{\text{self}}(8, 1, 3, 1)$ | O:1s $\times$ O:2p $\rightarrow \mathbb{3}$ | 1 | 1 |
| $I^{\text{self}}(8, 2, 3, 1)$ | O:2s $\times$ O:2p $\rightarrow \mathbb{3}$ | 1 | 2 |
| $I^{\text{self}}(8, 3, 1, 1)$ | O:2p $\times$ O:1s $\rightarrow \mathbb{3}$ | 1 | 3 |
| $I^{\text{self}}(8, 3, 2, 1)$ | O:2p $\times$ O:2s $\rightarrow \mathbb{3}$ | 1 | 4 |
| $I^{\text{self}}(8, 3, 3, 1)$ | O:2p $\times$ O:2p $\rightarrow \mathbb{3}$ | 1 | 5 |
| $I^{\text{self}}(8, 3, 3, 2)$ | O:2p $\times$ O:2p $\rightarrow \mathbb{5}$ | 2 | 1 |
| **irreps for off-diagonal blocks** | | | |
| $I^{\text{pair}}(Z_1, Z_2, n_1, n_2, L)$ | orbital interaction | degree | index |
| $I^{\text{pair}}(8, 1, 1, 1, 0)$ | O:1s $\times$ H:1s $\rightarrow \mathbb{1}$ | 0 | 1 |
| $I^{\text{pair}}(8, 1, 2, 1, 0)$ | O:2s $\times$ H:1s $\rightarrow \mathbb{1}$ | 0 | 2 |
| $I^{\text{pair}}(1, 8, 1, 1, 0)$ | H:1s $\times$ O:1s $\rightarrow \mathbb{1}$ | 0 | 3 |
| $I^{\text{pair}}(1, 8, 1, 2, 0)$ | H:1s $\times$ O:2s $\rightarrow \mathbb{1}$ | 0 | 4 |
| $I^{\text{pair}}(1, 1, 1, 1, 0)$ | H:1s $\times$ H:1s $\rightarrow \mathbb{1}$ | 0 | 5 |
| $I^{\text{pair}}(8, 1, 3, 1, 0)$ | O:2p $\times$ H:1s $\rightarrow \mathbb{3}$ | 1 | 1 |
| $I^{\text{pair}}(1, 8, 1, 3, 0)$ | H:1s $\times$ O:2p $\rightarrow \mathbb{3}$ | 1 | 2 |

Table S2: Breakdown of all irreps necessary to construct the Hamiltonian matrix for a water molecule using a minimal basis set. Here, we label the 1s orbital of H with the number 1, and the 1s, 2s, and 2p orbitals of O with the numbers 1, 2, and 3. Note that channel indices can be chosen arbitrarily, as long as indices for irreps of the same degree are unique and the assignment is consistent.

## C.3   Overlap matrix prediction

Since the overlap matrix $\mathbf{S}$ represents the overlap integral of atomic orbital basis functions, it is not affected by many-body effects and its entries depend only on the types of interacting elements and their relative orientation and distance. Thus, simplified feature representations can be used to construct it. Self-interaction features for the overlap matrix are obtained by passing the initial atomic

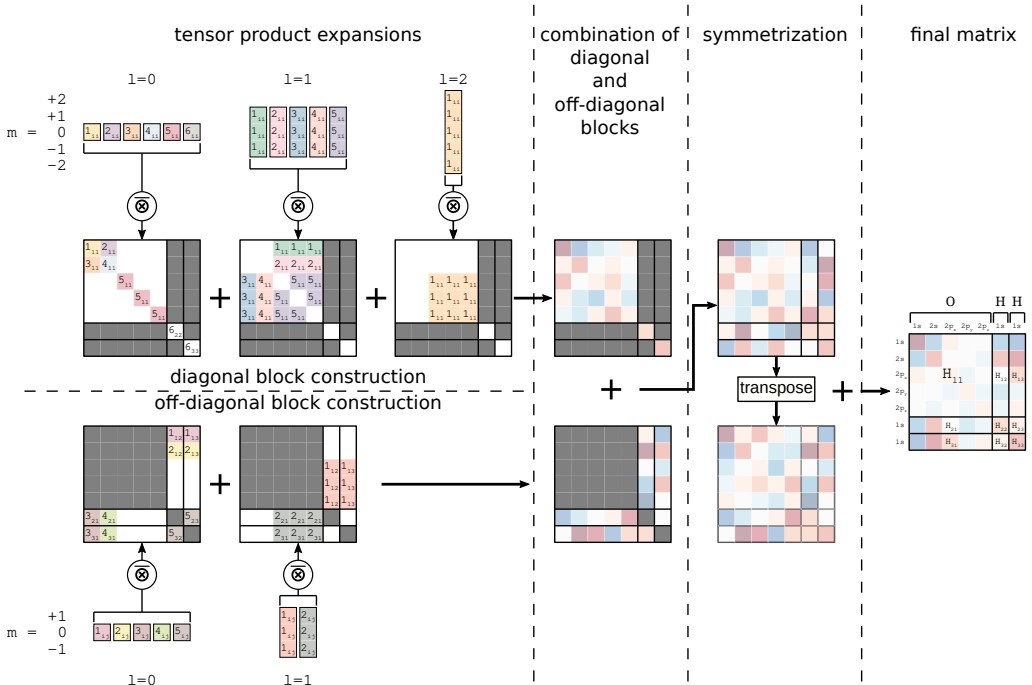

Figure S5: An illustration for the construction of the Hamiltonian matrix for a water molecule using a minimal basis set, showing how irreps of different degrees are transformed via tensor product expansions to reconstruct the Hamiltonian matrix block-by-block. The irreps are represented by colored squares labelled $c_{ii}$ or $c_{ij}$, where $c$ is the channel index (see Tab S2) and the subscript specifies atom indices. Different orders of an irrep are represented by individual squares of the same color. Expanded irreps are placed at the appropriate position in the matrix depending on the orbital-pair interaction that the block represents. The individual blocks are then added together to form the diagonal and off-diagonal parts of the matrix, after which the two parts are combined and the result is added to its transpose in order to ensure the final result is a symmetric matrix.

feature representations $\mathbf{x}$ (produced by the embedding layer from the nuclear charges $Z$) through a spherical linear layer and a residual block:

$$
\begin{aligned}
\mathbf{u}_{ii}^{(l)} &= \mathrm{sphlinear}_{L \to L, F \to F}(\mathbf{x}_i)^{(l)} , \\
\mathbf{s}_{ii} &= \mathrm{residual}(\mathbf{u}_{ii}) .
\end{aligned}
\tag{S10}
$$

Similarly, pair-interaction features are generated according to:

$$
\begin{aligned}
\mathbf{u}_{ij}^{(l)} &= \mathrm{pairmix}_{L, L \to L}(\mathbf{x}_i, \mathrm{sphlinear}_{L \to L, 1 \to F}\left(\mathbf{Y}(\mathbf{r}_{ij})\right), \|\mathbf{r}_{ij}\|)^{(l)} , \\
\mathbf{s}_{ij} &= \mathrm{residual}(\mathbf{u}_{ij}) .
\end{aligned}
\tag{S11}
$$

The overlap matrix is then assembled from the features $\mathbf{s}_{ii}$ and $\mathbf{s}_{ij}$ (insteaf of $\mathbf{f}_{ii}$ and $\mathbf{f}_{ij}$) analogously to other Hamiltonian matrices.

## D Ablation studies

To explore the impact of different possible architectural simplifications on the prediction accuracy of PhiSNet, the following ablated variants are considered:

- **shared matrix features**: The same features are used for central and neighboring atoms for the Hamiltonian matrix prediction. In other words, the parameters of $\mathrm{residual}$ in Eq. 16 are shared between the computation of $\mathbf{c}$ and $\mathbf{n}$, i.e. $\mathbf{c} = \mathbf{n}$.

- **shared interaction features**: The same features are used for central and neighboring atoms in interaction blocks (Eq. 13), i.e. $\mathbf{c} = \mathbf{n}$. In other words, the parameters of $\mathrm{sphlinear}$ and $\mathrm{residual}$ operations are shared between the computation of the $\mathbf{i}$ and $\mathbf{j}$ features in Eq. 15.

- **no selfmix**: All selfmix (Eq. 9) operations are removed from spherical linear layers (Eq. 10) when $L_{\text{in}} = L_{\text{out}}$, i.e.

$$\text{sphlinear}_{L \to L, F_{\text{in}} \to F_{\text{out}}}(\mathbf{x}) = \text{linear}_{F_{\text{in}} \to F_{\text{out}}}(\mathbf{x}) \ .$$

- **simple pair features**: No interaction with neighboring atoms is used for constructing self-interaction ($\mathbf{f}_{ii}$, Eq. 17) and pair-interaction ($\mathbf{f}_{ij}$, Eq. 18) features, i.e. they are computed as

$$\mathbf{f}_{ii}^{(l)} = \text{residual}\left(\mathbf{c}_i^{(l)}\right) \ ,$$

$$\mathbf{f}_{ij}^{(l)} = \text{residual}\left(\text{pairmix}(\mathbf{c}_i, \mathbf{c}_j, \|\mathbf{r}_{ij}\|)^{(l)}\right) \ .$$

The results of the ablation studies are shown in Table S3. We find that all considered modifications significantly reduce the prediction accuracy of PhiSNet. With exception of the "no selfmix" variant, any improvements in computational efficiency obtained by simplifying the architecture are negligible ($<1\%$) and do not justify the observed reduction in accuracy. The "no selfmix" variant, however, although between 14-77% less accurate than the baseline model, can be a valid alternative for some applications: We find that inference speed and training times are reduced by roughly 50% for this variant.

Table S3: Prediction errors (in units of $10^{-6}$ $\text{E}_{\text{h}}$) for Kohn-Sham matrices of ablated variants compared with the baseline architecture. Note that models are not trained to convergence, but for a fixed number of training steps (40k steps for models trained on water, 30k steps for models trained on ethanol).

| dataset | baseline | shared mat. feat. | shared int. feat. | no selfmix | simple pair feat. |
|---------|----------|-------------------|-------------------|------------|-------------------|
| Water   | 19.18    | 24.93             | 29.57             | 33.88      | 88.21             |
| Ethanol | 77.29    | 81.70             | 89.47             | 87.85      | 121.63            |

# E   Speedup with respect to *ab initio* calculations

## E.1   Model inference times compared to *ab initio* calculations

To evaluate the speedup PhiSNet provides compared to *ab initio* calculations, we compare model inference times to DFT calculations at the PBE/def2-SVP level of theory with the ORCA 4.0.1 software [28, 29]. PhiSNet was evaluated on an NVIDIA A100 GPU with a batch size of 64, whereas DFT calculations were performed on Intel Xeon E5-2690 CPUs. The average wall-clock times for evaluating a single structure are summarized in Tab. S4.

| molecule | DFT | PhiSNet | speedup |
|----------|-----|---------|---------|
| ethanol | 21.657 s | 0.027 s | $\sim 802\times$ |
| malondialdehyde | 38.923 s | 0.029 s | $\sim 1342\times$ |
| uracil | 86.675 s | 0.050 s | $\sim 1734\times$ |
| aspirin | 343.615 s | 0.155 s | $\sim 2217\times$ |

Table S4: Average wall-clock time for evaluating a single structure with DFT and PhiSNet for different molecules.

## E.2   Speeding up *ab initio* calculations using predicted wavefunctions as initial guess

Although PhiSNet achieves low prediction errors with respect to quantities calculated from *ab initio* calculations, even small errors might be unacceptable for applications that require exact solutions. Even then, PhiSNet can speedup quantum chemistry by providing the predicted wavefunctions as initial guess (typically, the guess is obtained from a semi-empirical method such as extended Hückel theory [30] or from using model potentials). Tab. S5 summarizes the reduction in iterations

until convergence and total wall-clock time for different SCF algorithms when using wavefunctions predicted by PhiSNet instead of the default guess. Note that the quality of guess wavefunctions provided by PhiSNet also makes it possible to use quadratically converging SCF algorithms like SOSCF and NRSCF instead of the standard DIIS approach. All DFT calculations were performed with the ORCA 4.0.1 software [28, 29].

| molecule | # iterations until convergence | | | total wall-clock time | | |
|---|---|---|---|---|---|---|
| SCF algorithm | DIIS | SOSCF | NRSCF | DIIS | SOSCF | NRSCF |
| **ethanol** | 47.1% | 44.6% | 68.6% | 31.5% | 30.4% | 36.5% |
| **malondialdehyde** | 51.3% | 42.4% | 71.5% | 40.5% | 34.3% | 40.0% |
| **uracil** | 47.3% | 40.4% | 67.1% | 38.1% | 32.9% | 38.0% |
| **aspirin** | 33.4% | 29.5% | 55.6% | 29.1% | 24.8% | 24.5% |

Table S5: Average reduction in the number of iterations until convergence and total wall-clock time for different molecules and SCF algorithms when starting from the default versus PhiSNet-predicted guess wavefunction.

## F  Datasets

The DFT datasets (containing energies, forces, Kohn-Sham matrices and overlap matrices) for water, ethanol, malondialdehyde and uracil were taken from [31] and are also available on http://quantum-machine.org/datasets/. The corresponding data for the aspirin molecule was obtained from [32]. With the exception of water, these datasets are based on subsets of structures drawn from the MD17 dataset. Water structures were sampled using a classical force field. All quantities in these datasets have been computed at the PBE/def2-SVP level of theory, where PBE denotes the computational method (in this case density functional) and def2-SVP the atomic basis set used for expanding the wavefunction. The water datasets used in the transfer learning experiments use the same molecular structures as the DFT water data. Based on these structures energies, forces, as well as Fock, overlap and core Hamiltonian matrices (the latter three only in the case of HF/cc-pVDZ) were computed at the HF/cc-pVDZ and CCSD(T)/cc-pVTZ levels of theory using the PSI4 code package [33]. Further details, such as data set size, are provided in Table S6.

Table S6: Datasets used in this work. The level of theory refers to the combination of method and basis set used to calculate molecular properties. $N_{\mathrm{mol}}$ denotes the number of molecular structures in each dataset, while $N_{\mathrm{atoms}}$ is the number of atoms. $N_{\mathrm{basis}}$ is the total number of atomic basis functions used to express the wavefunction of a molecule. All matrix properties (Fock matrix, core Hamiltonian, Kohn-Sham matrix and overlap matrix) are of the dimension $N_{\mathrm{basis}} \times N_{\mathrm{basis}}$. We omit this value for the last row, where no matrix quantities have been computed.

| Dataset | Level of theory | $N_{\mathrm{mol}}$ | $N_{\mathrm{atoms}}$ | $N_{\mathrm{basis}}$ | Source |
|---|---|---|---|---|---|
| Water | PBE/def2-SVP | 4 999 | 3 | 24 | [31] |
| Ethanol | PBE/def2-SVP | 30 000 | 9 | 72 | [31] |
| Malondialdehyde | PBE/def2-SVP | 26 978 | 9 | 90 | [31] |
| Uracil | PBE/def2-SVP | 30 000 | 12 | 132 | [31] |
| Aspirin | PBE/def2-SVP | 30 000 | 21 | 222 | [32] |
| Water | HF/cc-pVDZ | 4 999 | 3 | 24 | this work |
| Water | CCSD(T)/cc-pVTZ | 4 999 | 3 | - | this work |

## G  Training procedure and hyperparameters

All models in this work use $F = 128$ feature channels with a maximum degree $L_{\mathrm{max}} = 4$, and $K = 128$ basis functions with a cutoff radius of $r_{\mathrm{cut}} = 15\ a_0$ (see Eq. S9), resulting in roughly 17M parameters (for comparison, SchNOrb [31] uses 93M parameters). The parameters were optimized with AMSGrad [34] using an initial learning rate of $10^{-3}$, other hyperparameters of the optimizer were set to the recommended defaults. The performance on the validation set was evaluated every 1k

training steps and the learning rate decayed by a factor of 0.5 if the validation loss did not decrease for 10 consecutive evaluations. Training was stopped once the learning rate was smaller than $10^{-5}$ and the best-performing model (lowest validation loss) was selected. All models were trained on a single NVIDIA Titan Xp GPU.

## G.1 DFT datasets

Models for DFT datasets were trained by minimizing the loss function

$$\mathcal{L} = \frac{1}{N_{\text{batch}}} \sum_{b=1}^{N_{\text{batch}}} \left( \|\mathbf{K}_b^{\text{ref}} - \mathbf{K}_b\|_F^2 + \|\mathbf{S}_b^{\text{ref}} - \mathbf{S}_b\|_F^2 \right) , \tag{S12}$$

where $\mathbf{K}_b$ and $\mathbf{S}_b$ are the predicted Kohn-Sham and overlap matrices for structure $b$ in the batch, $\mathbf{K}_b^{\text{ref}}$ and $\mathbf{S}_b^{\text{ref}}$ denote the corresponding reference matrices, and $\|\cdot\|_F^2$ is the squared Frobenius norm. The batch, training set, validation set, and test set sizes used for the different datasets are given in Tab. S7.

Table S7: Batch, training set, validation set, and test set sizes for models trained on the DFT datasets. Note that batch sizes between different molecules were not tuned for performance. All variations are due to different sized Hamiltonian matrices and the memory constraints of the training hardware (e.g. water consists of 3 atoms and Hamiltonian matrices are $24 \times 24$, whereas aspirin consists of 21 atoms and Hamiltonian matrices are $222 \times 222$).

| Dataset | $N_{\text{batch}}$ | $N_{\text{train}}$ | $N_{\text{valid}}$ | $N_{\text{test}}$ |
|---|---|---|---|---|
| Water | 50 | 500 | 500 | 3 999 |
| Ethanol | 10 | 25 000 | 500 | 4 500 |
| Malondialdehyde | 10 | 25 000 | 500 | 1 478 |
| Uracil | 5 | 25 000 | 500 | 4 500 |
| Aspirin | 2 | 25 000 | 500 | 4 500 |

## G.2 Transfer learning from HF to CCSD(T)

For the transfer learning experiment, the model is trained on the same structures with data computed both at the HF/cc-pVDZ and CCSD(T)/cc-pVTZ levels of theory (see last two rows of Tab. S6). We use a batch size of $N_{\text{batch}} = 25$, and training, validation, and test sets of sizes $N_{\text{train}} = 500$, $N_{\text{valid}} = 500$, and $N_{\text{test}} = 3\,999$.

First, the model is trained by minimizing the loss function

$$\mathcal{L}_1 = \frac{1}{N_{\text{batch}}} \sum_{b=1}^{N_{\text{batch}}} \left( \|\mathbf{F}_b^{\text{ref}} - \mathbf{F}_b\|_F^2 + \| + \|\mathbf{H}_b^{\text{core,ref}} - \mathbf{H}_b^{\text{core}}\|_F^2 + \|\mathbf{S}_b^{\text{ref}} - \mathbf{S}_b\|_F^2 \right) , \tag{S13}$$

where $\mathbf{F}_b$, $\mathbf{H}_b^{\text{core}}$, $\mathbf{S}_b$ are the predicted Fock, core Hamiltonian, and overlap matrices for structure $b$ in the batch, and $\mathbf{F}_b^{\text{ref}}$, $\mathbf{H}_b^{\text{core,ref}}$, $\mathbf{S}_b^{\text{ref}}$ denote the corresponding HF-level reference matrices. Although a model trained in this way reaches matrix prediction errors that are comparable with the results obtained for DFT datasets (see Tab. 5), we observe that errors for energies and forces derived from the predicted matrices (using Eq. S4) are much larger. We speculate that this is due to the fact that all matrix elements are weighed equally when computing $\mathcal{L}_1$, although some elements will have a much greater effect on the energy and forces than others. For this reason, we re-train our model on the HF data using a new loss function $\mathcal{L} = \mathcal{L}_1 + \mathcal{L}_2$ with

$$\mathcal{L}_2 = \frac{1}{N_{\text{batch}}} \sum_{b=1}^{N_{\text{batch}}} \left( (E_b^{\text{ref}} - E_b)^2 + \frac{1}{N} \sum_{i=1}^{N} \left\| -\frac{\partial E_b}{\partial \mathbf{R}_{b,i}} - \mathbf{f}_{b,i}^{\text{ref}} \right\|^2 \right) , \tag{S14}$$

which incorporates information from HF-level energy and forces directly. Here, $E_b$ is the predicted energy for structure $b$ in the batch, $E_b^{\text{ref}}$ the corresponding HF reference, $\mathbf{f}_{b,i}^{\text{ref}}$ the reference force acting on atom $i$ in structure $b$ and $\mathbf{R}_{b,i}$ its Cartesian coordinates (in total, each structure contains $N$ atoms). Force predictions are obtained using automatic differentiation. After re-training with the modified loss, energy and force predictions are improved substantially, although the prediction of

| model | **F** $[10^{-6}\ \mathrm{E_h}]$ | **S** $[10^{-6}]$ | $\mathbf{H}_{\mathrm{core}}$ $[10^{-6}\ \mathrm{E_h}]$ | energy $[10^{-3}\ \mathrm{E_h}]$ | forces $[10^{-3}\ \mathrm{E_h}/a_0]$ |
|---|---|---|---|---|---|
| trained on $\mathcal{L}_1$ | 24.77 | 1.06 | 39.90 | 6.51 | 57.10 |
| re-trained on $\mathcal{L}_1 + \mathcal{L}_2$ | 46.94 | 5.32 | 58.00 | 0.117 | 2.289 |

Table S8: Comparison of mean absolute prediction errors for models trained on HF/cc-pVDZ reference data.

matrix elements deteriorates slightly (see Tab. S8). We also experimented with training a model directly on $\mathcal{L} = \mathcal{L}_1 + \mathcal{L}_2$, but found that this does not work: The matrices predicted at the beginning of training lead to numerical instabilities when solving the generalized eigenvalue problem (Eq. S3), i.e. a warm-up-phase using the pure matrix loss $\mathcal{L}_1$ is necessary for convergence.

Finally, the re-trained model is fine-tuned using the loss function

$$\mathcal{L}_{\mathrm{TL}} = \frac{1}{N_{\mathrm{batch}}} \sum_{b=1}^{N_{\mathrm{batch}}} \left( \frac{1}{N} \sum_{i=1}^{N} \left\| -\frac{\partial E_b}{\partial \mathbf{R}_{b,i}} - \mathbf{f}_{b,i}^{\mathrm{ref,CC}} \right\|^2 + \|\mathbf{H}_b^{\mathrm{core,ref}} - \mathbf{H}_b^{\mathrm{core}}\|_F^2 + \|\mathbf{S}_b^{\mathrm{ref}} - \mathbf{S}_b\|_F^2 \right) \tag{S15}$$

where $\mathbf{f}_{b,i}^{\mathrm{ref,CC}}$ are the reference forces computed at the CCSD(T)/cc-pVTZ level of theory, whereas the reference matrices $\mathbf{H}_b^{\mathrm{core,ref}}$ and $\mathbf{S}_b^{\mathrm{ref}}$ are taken from the HF/cc-pVDZ data. Note that there is a constant energy offset between CCSD(T)/cc-pVTZ and HF/cc-pVDZ data, which cannot be expressed when deriving energies with Eq. S4 without also modifying the core Hamiltonian, which is why CCSD(T)/cc-pVTZ level reference energies are not used in Eq. S15. For any practical chemical application, only the relative energy between two structures is a well-defined quantity (absolute energies can be modified arbitrarily by adding constants without changing the underlying physics). To still be able to compute both energy and force errors with respect to the CCSD(T)/cc-pVTZ reference (see Section 5 in the main text), we analytically fit an energy shift constant that minimizes the squared error between energy predictions and reference energies in the training set and add it to the model prediction before evaluating it on the test set.