# OpenReview forum: "SE(3)-equivariant prediction of molecular wavefunctions and electronic densities"
_NeurIPS.cc/2021/Conference — NeurIPS 2021 Poster_

### Official Review · Reviewer_1y2Z · 2021-07-11

**Rating:** 7
**Confidence:** 2

**Summary:**

The paper proposes PhiSNet, an SE3 equivariant NN whose inputs are distance vectors between atoms of a given molecule with nuclear charges, and outputs a prediction of the entries of the Hamiltonian matrix that is used to solve the Schrodinger equation to find the electronic wavefunction of the molecule. It is claimed that training PhiSNet on a molecular dataset reaches state of the art prediction accuracies on various molecules.


**Main Review:**

Without a background in quantum chemistry, it is difficult for me to assess the significance of the results in the paper, hence I will focus on assessing the proposed methodology.

The paper attributes the success of PhiSNet over the baseline SchNOrb to its SE3 equivariance, but it is unclear how much of the improvements over the baseline are due to its equivariance and how much is from design choices of the network. Of course these two come hand in hand, so it's difficult to separate their contributions to improving prediction accuracy, but a more detailed description of the baseline method along with a description of the difference between this and PhiSNet would be desirable. In addition, given the claims of SOTA, it would be useful to have ablations to explain the various design choices of PhiSNet, e.g. parameterisation of Interaction blocks (eq 13), using a different set of parameters for the central point features and the neighbouring point features (eq 15, 16), or using the other neighbours of atom i for computing the pair interaction features between atoms i & j (eq 18). In summary, a better breakdown of what design choices have led to how much improvement in prediction accuracy would be very helpful for explaining the success of PhiSNet and will encourage follow up work to make further improvements.

The paper introduces PhiSNet as a novel SE3 equivariant architecture for point cloud data, but it's not clear where the novelty is: it's not clear whether the authors are claiming some of the building blocks in Section 4.1 are novel or that the combination of these existing building blocks is novel. The building blocks described in Section 4.1 are lacking in citations e.g. Tensor product contractions have been used frequently in the SE3 equivariance literature, so relevant work should be cited e.g. Clebsch-Gordan Nets [Kondor et al, 2018].

I appreciate the paper's attempt to make the methodology clear, with details on notation and the methodology, but there remain some points of confusion
- In line 207 it is said 'tilde{x} serves as input to the next module in the chain' but I don't see tilde{x} appearing subsequently.
- In line 209, what is the sum across?
- I find the use of Tensor product expansions to construct matrix predictions interesting, but the explanation in Section 4.2 is not so clear (e.g. what is used as 'a' in eq 15?). It would be helpful to include Figure S5 as part of the main paper for this.


**Time Spent Reviewing:**

4

---

> ### Author Response · Authors · 2021-08-09
> **Response to Reviewer 1y2Z**
>
> Thank you for suggesting ablation studies! We agree that they are helpful to assess the impact of different architectural components. Based on your suggestions, we tested the following variants:
> - shared matrix features: The same features are used for central and neighboring atoms for the Hamiltonian matrix prediction (see Eq. 16, errors increase between 6-30%).
> - shared interaction features: The same features are used for central and neighboring atoms in interaction blocks (see Eq. 13, errors increase between 16-54%).
> - no selfmix: All selfmix (Eq. 9) operations are removed from spherical linear layers (Eq. 10) when the input and output degree $L$ is equal (errors increase between 14-77%).
> - simple pair features: No interaction with neighboring atoms is used for constructing self-interaction (Eq. 17) and pair-interaction (Eq. 18) features (errors increase between 57-360%).
>
> More details and a discussion of these results will be added to the supporting information for the camera-ready version.
> As you can see, the combination of building blocks used here does indeed lead to the best performance. In addition, more details on the differences between SchNOrb and PhiSNet will be added to the text.
>
> Regarding the difference between SchNOrb and PhiSNet due to the SE(3)-equivariance, one could say that introducing equivariance is an important design choice. The benefits in doing so are perhaps best reflected in the data efficiency of both methods. The SchNOrb architecture relies on data augmentation for training in order to learn (approximate) equivariant behavior. While this works reasonably well for small structures, it becomes increasingly inefficient for larger molecules due to the complexity of the Hamiltonian matrices, as can be seen using aspirin as an example. Unfortunately, this effect can not be mitigated by increasing the number of network parameters, as it is a fundamental problem of not capturing the necessary symmetries in the data. PhiSNet on the other hand does not require data augmentation at all and at the same time consistently yields better predictions using fewer parameters.
>
> The novelty of PhiSNet is mainly in the tensor product expansion, the combination of the individual layers, and in the way the Hamiltonian matrix is constructed from irreducible representations. We will rewrite Section 4.1 in the camera-ready version to make clearer which parts are novel and include a reference to [Kondor et al, 2018].
>
> Thank you for pointing out sections of the paper which are confusing. We hope we are able to address the points you mentioned adequately:
> - The model is composed of multiple modules of the form described in Eq. 15, so it is meant that $\tilde{\mathbf{x}}$ is used as the input $\mathbf{x}$ in the next such module. We will rephrase the text to make this clearer (see also Figure S2 in the supplement for an illustration of the module structure and the role of $\mathbf{x}$).
> - The sum is across outputs $\mathbf{y}$ of each module in the atomic feature representation network. This will be stated more clearly in the text (please also see Figure S2 for a clarifying illustration).
> - The values for $\mathbf{a}$ are extracted from the self- and pair-interaction features using the indexing process described in lines 229-236. We are aware that the process is complex and hard to follow and will try to improve the clarity of the description. We would have liked to include Figure S5 and a much more detailed explanation in the main text, but unfortunately could not do so due to the page restriction.
>
> If you feel the additions (i.e. ablation studies) we want to include for the camera-ready version improve the quality of our manuscript, please consider updating your rating.

---

> > ### Comment · Reviewer_1y2Z · 2021-08-19
> > **Response to rebuttal**
> >
> > Thank you for the thorough rebuttal. The rebuttal has addressed my concerns and so I'll update my rating accordingly.

---

### Official Review · Reviewer_ZUAT · 2021-07-14

**Rating:** 8
**Confidence:** 4

**Summary:**

The paper describes a method to predict the wavefunctions and electron densities using a rotationally equivariant method. The method achieves sufficient accuracy for the model to be used to predict properties such as energies and forces directly from atom positions and charges. The approach is evaluated by comparing to an existing rotationally-invariant approach using prediction errors in Kohn-Sham matrices, energies and wavefunction cosine similarity as metrics to compare the methods. Background on relevance and significance as well as implementation details are provided in the supplementary information along with sample code.

**Limitations And Societal Impact:**

Yes, to a reasonable extent.

**Main Review:**

**Main paper**

1. Related works: The following highly related references should be cited since they are part of very related threads of work:

   [1] "SE(3)-Transformers: 3D Roto-Translation Equivariant Attention Networks" by Fabian B. Fuchs and Daniel E. Worrall and Volker Fischer and Max Welling

   [2] "OrbNet: Deep learning for quantum chemistry using symmetry-adapted atomic-orbital features" by Zhuoran Qiao, Matthew Welborn, Animashree Anandkumar, Frederick R. Manby, and  Thomas F. Miller

   At the very least some comments on how the proposed method differs from these approaches should be included. Reference [1] presents an equivariant network for property prediction and reference [2] includes a 1000x speedup over DFT by leveraging symmetry-aware features for orbital data.

2. The abstract, Section 1 and Section 3 (theory development) all mention that the underlying geometries of molecules need to to rotation-equivariant to capture the various rotational symmetries without losing the information about the rotation. This suggests that an SO(3)-equivariant network would be sufficient. However, there is an abrupt jump to SE(3)-equivariance in Section 4 (Deep learning architecture). What is the reasoning behind that? Note that SE(3) expresses $[R | t]$ and SO(3) expresses $R$, where $R$ is the rotation matrix and $t$ is the translation vector in $\mathbb{R}^3$. We can ignore the homogeneous formulation's scale factor for now. How/When do translations need to be preserved and predicted? How do translations even manifest in a representation of a molecule? Do individual bonds within a molecule elongate or contract (in response to temperature or pressure, for example) and if so, then how is that data fed into the network? Or is SE(3)-equivariance being used as a superset of SO(3)? Or is it the case that properties of groups of atoms within a molecule are of interest, and the bonds attaching these groups to rest of the molecule can change length, thereby encoding some useful information in the translation components?

3. Figure 1 (D). The HF-level function, the correction (amplified) and the CC-level function are presented. By eye, the HF-level function and CC-level function look the same to me. Is it correct to expect 'CC-level function = HF-level function + correction'? If not, how should we interpret Figure 1-D? A clarifying caption would be helpful. Alternatively, the differences in HF-level function and CC-level function could be highlighted.

4. Evaluation [Table 1]: The paper clearly states that the approach is built on PhySNet, yet the proposed method (PhiSNet) is not compared to PhySNet. It is instead compared only to SchNOrb. The results presented in Table 1 for SchNOrb do correlate with the results in the supplementary material of the SchNOrb publication, but only partially. So I presume authors ran additional experiments. Great.

5. Evaluation [Table 1]: It would be good to have a comparison to PhySNet here as it would allow comparison to a whole host of methods. OrbNet evaluates a number of methods against DFT including PhySNet and at a 1000x speedup on DFT. It would be good to get a rough estimate of speedup of the proposed method wrt DFT as a general guideline.

6. Evaluation: Several other works in this area present results for property prediction (QM9, QM7, DrugBank etc). It is curious to not find these included in the paper as the paper explicitly claims to provide good transferability to property prediction. I do note that some results wrt HF to CCSD transfer learning are provided in the supplementary material but as these measure very different quantities from those in related works, it's unclear how to interpret the results. (Apologies, I am still getting my head round the level of theory business).

7. Section 4.1 is missing a sentence or two in way of introduction. It's quite unclear whether the layers defined in Section 4.1 refer to PhiSNet (this paper) or PhySNet (a previous work upon which this paper builds). I am assuming they refer to PhiSNet (this paper) based on the cited references inline and line 144-145, but just staing this upfront would be easier for the reader.

8. Section 4.1 [Interaction blocks]: This is the first point where positions $\mathbf{r}$ of the atoms and the inter-atom distances $\mathbf{r}_{ij}$ are used. So coming back to my earlier point about SO(3) vs SE(3), there could be an argument here for needing SE(3) to encode the bond lengths between the atoms. Much as before, aren't these known and fixed quantities or do they vary substantially, and if they do, do datasets actually have that variation? Else, why not just assume some units of 'one C-H bond away'? (Of course, things get complicated when the bonds are not unique, i.e. valence allows either double or triple bond between C and N etc). But this can largely by disambiguated by using valence constraints.

9. In Eq (16), it might be good to add some suffix or words to clarify that the $\mathbf{f}$ in the definition of $\mathbf{c} = residual(\mathbf{f})$ and $\mathbf{n} = residual(\mathbf{f})$ operates on different atoms within the expressions itself.

10. Network size: A comparison of the number of parameters is provided wrt SchNOrb in the supplementary information (17M for PhiSNet and 34M for SchNOrb). How many parameters does PhySNet have, seeing as that's the network this paper is most closely based on?

11. In the contributions section, the authors say the proposed method improves over state of the art by two orders of magnitude. It would be helpful to clarify whether this is the case when comparing to equivariant methods or non-equivariant methods? I presume it is the latter. Going back to the point on evaluation, as the main evaluation only compares against one other method, it would be good to add a comparison to PhySNet and perhaps also, where possible a partial/alternative comparison to other methods, such as the ones OrbNet uses for evaluation with a accuracy-effort tradeoff to highlight the benefits of the current approach (end-end trainable). Ballpark number could also suffice for such evaluation.

12. There is a slight contradiction in the contributions section (Lines 49-53): Contribution 1 suggests the need for SE(3) whereas contribution 2 suggests only transformation under rotations are required to be preserved.

13. Evaluation: PhySNet seems to improve with ensembles. Does it help with PhiSNet? If it's not needed, that could be a good thing to highlight.

14. Would the method scale to work on proteins (larger molecules but more repeated structures) as it does for small molecules. Most of the examples included in the paper are for small molecules. There is a passing comment about computational complexity scaling with the number of atoms in Section 5, but my question is more around the repeated structures aspect.


**Supplementary information:**

1. Thank you! The background information on quantum chemistry (Section A) leading up to the embedding details (Section C) along with the network architecture diagrams and Hamiltonian matrix construstion with minimal basis sets (Section D and Figure S5) are the clearest I have seen in this field. To make this even better, I would suggest using terminology from both ML/CV and Physics where relevant. Group equivariance is heavily studied in both physics and computer vision, for example, but different terminology is used. Perhaps for electronic dissemination, it would make sense to have a long version of this paper that interweaves the main paper and the supplementary info so as to give the reader an uninterrupted (if longer) and lucid flow of reading.

2. Why do the batch sizes used for different molecules (Table S4) vary so much? Is it in direct proportion to the number of atoms per molecule? Do you not have to change the learning rate or the number of epochs to adapt? In other words, are the number of atoms within a batch for the different molecules roughly constant?

**Notes**

Writing style: At several points while reading the paper, I had questions along the lines of 'What do the different levels that the authors refer to mean?' or 'What is **S** matrix'? All these questions are answered nicely in the supplementary information, but then it's unclear what order to read the paper and supplementary info in? Given the topic of the paper, quite a bit of background from quantum chemistry/physics is required, I think it's fine for there to be a substantial supplementary information for those who may need it, but it would be nicer for all readers if the supplementary info was referred to in the main paper whenever the background from supplementary info is required for understanding the content in the main paper. For example, 'We predict **S** (see Supp. info Section .. for explanation of **S**)'.

**Overall assessment**

This paper presents a very good motivation of DL methods for quantum chemistry,specifically learning wavefunctions from electronic orbital information. The method presented is novel even though it builds on previous work in the area. The method addresses a key challenge and demonstrates significant improvement in accuracy of wavefunction and energy prediction with an end-end approach. However, some comparisons and comments on differences wrt key existing works would be helpful in understanding the importance of this work.

**Time Spent Reviewing:**

6

---

> ### Author Response · Authors · 2021-08-09
> **Response to Reviewer ZUAT**
>
> **Main paper**
> 1. We will include these references and also discuss SE(3)-Transformers and OrbNet explicitly. The main difference of our model compared to most previous models comes from the fact that it is able to predict molecular wavefunctions, which is much more complex than the direct prediction of molecular properties, as predicting wavefunctions gives access to the full electronic information. The advantage is that a vast range of quantum mechanical properties can be derived once the wavefunction is known, whereas others models need to be re-trained for every property of interest.
> For example, properties which can be derived from wavefunctions include the orbital features which OrbNet relies on (as inputs) for its predictions, which otherwise need to be derived from (comparatively more expensive) quantum mechanical calculations.
> 2. Yes, the individual bonds of molecules constantly elongate or contract due to thermal motion (even at absolute zero, bond lengths fluctuate due to the zero point energy). Even small changes in bond lengths typically change the entries of Hamiltonian matrices appreciably, so it is important that a model responds correctly to relative translations of atoms. However, it is crucial that model predictions only change with respect to relative translations. Overall translations of all atoms on the other hand should lead to the same output. We will rephrase the text to clarify what is meant by SE(3)-equivariance in this context and why it is important.
> 3. You are correct, the 'CC-level function' is equal to 'HF-level function + correction'. However the magnitude of the correction is much smaller compared to that of the HF-level function, which is why it is shown in an amplified manner. Due to the low magnitude of the correction, the difference between the HF-level and CC-level function cannot be spotted by eye.
> This will be clarified in more detail in the caption of Figure 1-D. Interestingly (and perhaps surprisingly), even minor differences have a significant effect on the resulting properties, as demonstrated by our experiments in Section 5.
> 4. It is true that the architectural design of PhiSNet is based on PhysNet. However, PhysNet was designed to only predict certain properties such as energies and forces. Hence, it is not possible to predict molecular wavefunctions with PhysNet without significant and intellectually interesting modifications to its architecture. PhiSNet is exactly that: A heavily revamped PhysNet architecture capable of predicting wavefunctions. SchNOrb is the only model we are aware of, with which a direct comparison is possible. Your second assumption is correct, the authors of the SchNOrb publication were kind enough to provide some additional experiments and benchmarks for us.
> 5. The goal of PhiSNet is to predict the full molecular wavefunction of a chemical system. Methods like OrbNet and PhysNet on the other hand focus on predicting a set of individual properties and are unable to predict wavefunctions without major modifications. This is why a direct comparison to other methods (with the exception of SchNOrb) is unfortunately not possible (as they solve a different task). Regarding the speedup w.r.t. DFT: The DFT calculations for aspirin take about 344 s on average per data point. In contrast, on an A100 GPU, evaluating our model takes about 155 ms per data point, so we achieve similar speedups as OrbNet. A detailed table with timings for all molecules will be added to the supporting information.
> 6. As mentioned above, direct property prediction is a different (and simpler) task compared to predicting molecular wavefunctions (all quantum mechanical properties can be directly derived from the wavefunction from known physical relations). Unfortunately, existing benchmarks like QM9, QM7, or DrugBank do not contain the necessary reference data to train a model for predicting molecular wavefunctions. It is not possible to evaluate our model on these benchmarks without re-running all quantum mechanical calculations to derive the necessary training data. We agree that augmenting common benchmarks in this manner would be interesting, but is beyond the scope of the present work as it requires significant supercomputing time.
> 7. We thank the reviewer for pointing this out and will rephrase Section 4.1 to improve clarity.
> 8. Indeed, the bond lengths vary and this is also reflected in most datasets (unless they contain only equilibrium geometries). In fact, most of the challenge in the learning problem comes from capturing the complex ways in which the wavefunction changes when the bond lengths vary by small amounts. Note that PhiSNet only receives Cartesian coordinates as input (i.e. no "bonding information"). While bonds are certainly a very useful concept in chemistry, the interaction between atoms varies smoothly with distance and it is hard to define objectively at what point a "bond" between two atoms should be considered formed/broken.
> Things can get even more complicated and there are systems where bonding is quite ambiguous (e.g. transition metal complexes).
> The same holds true for valence and bond orders, especially in highly conjugated systems.
> A model that does not rely on hand-crafted "bonding information" is therefore preferable.
> 9. Thank you for the suggestion, we will rephrase the description of Eq. 16 for better clarity.
> 10. As stated in previous answers, while the architectural design of PhiSNet is inspired by PhysNet, a one to one comparison between both models is not meaningful, because they are designed for different applications (and contain very different building blocks). However, we understand that the text is confusing in this regard. We will rephrase it to make the connection (and differences) between PhysNet and our architecture more clear.
> 11. As stated in the answers to points 5 and 6, a comparison to methods like PhysNet and OrbNet is not directly possible, because they solve different tasks. To give a (crude) analogy, comparing PhiSNet to methods like OrbNet/PhysNet would be like comparing a model for image generation (PhiSNet) to models for image classification (on an image classification benchmark). The only other method (we are aware of) with which a direct comparison is meaningful is SchNOrb (a non-equivariant method).
> We understand that we did not state the differences to most other models clearly enough and will change the text accordingly.
> 12. Thank you for pointing this out, this paragraph will be rephrased more carefully.
> 13. It is likely that PhiSNet would also improve with ensembles. However, the improvement over the previous state of the art is already so substantial that we did not see the need to test this scenario, as further improvements would likely not be orders of magnitude.
> 14. The problem with large structures such as proteins is that it is very challenging to generate the reference data (computing just a single data point may take weeks to months, or even be completely unfeasible). However, exploiting repeated substructures in proteins to generate a model for a full protein is certainly an interesting concept. This idea should be explored in future work.
>
> **Supplementary information**
> 1. Thank you for the positive feedback and suggestions. Due to the page limit, we were unfortunately forced to shift a large part of the background information to the supplement so we could focus on the technical details and results in the main text. We agree that a longer background section in the main text would greatly improve clarity and hope we are able to fit more details into the main text for the camera-ready version.
> 2. Batch sizes for different molecules were chosen according to the memory constraints of the GPUs used for training (this will be explained in the revised manuscript). For example, water consists of 3 atoms and Hamiltonian matrices are $24 \times 24$, whereas aspirin consists of 21 atoms and Hamiltonian matrices are $222 \times 222$. Note that we did not tune batch sizes for performance and believe that our model is fairly robust to minor changes in the hyperparameters.
>
> **Notes**
>
> Thank you for these suggestions, we will add more references to the supplementary information at the appropriate locations in the main text to improve clarity.

---

> > ### Comment · Reviewer_ZUAT · 2021-08-21
> > **Re: Response to Review**
> >
> > As a person who does not have a background in chemistry, I found the clarifications (and the original paper) quite insightful. Most of my questions in the review were answered by the clarifications in author's response in Points 1 (difference in models wrt PhysNet and OrbNet) and 2 (Bond lengths vary, hence translations are important for wavefunction prediction, hence the use of SE3 rather than SO3). Clarifications to other points follow from these. I would encourage the authors to include the text used in the author response for Point1 and 2 in the main text for the benefit of other readers (and also to communicate the impact of the work fully to the community).
> > The experiments for proteins or the effect of ensembles would be an interesting follow-up, but I would agree that it doesn't need to be in this paper. The paper makes an important contribution as it stands. Following on from the clarifications (the main one being the use of translations owing to varying bond lengths), I am revising my rating to 8 (clear accept).

---

> > > ### Author Response · Authors · 2021-09-06
> > > **Re: Re: Response to Review**
> > >
> > > We agree that the clarifications included in our reply might be helpful for other readers and will follow your suggestion to include them in the manuscript for the camera-ready version. Thank you for your highly constructive and valuable review!

---

### Official Review · Reviewer_5CND · 2021-07-16

**Rating:** 6
**Confidence:** 3

**Summary:**


This contribution describes a method that approximates quantum mechanical total energies and wavefunctions with compute costs that are similar to those of Hartree Fock calculations. The paper describes a novel architecture, PhiSNet, that predicts wavefunctions from nuclear charges and positions: an equivariant featurization of the atomic information feeds a network that is built using a number SE(3) equivariant building blocks that in turn feeds to a block that builds the Hamiltonian matrix. The authors demonstrate that PhiSNet can train to predict DFT total energies to accuracies of about 0.11 kcal/mol for aspirin and better for smaller molecules. They further demonstrate that the network can "correct" a HF level wavefunction to an effective wavefunction that includes the effects of electron correlation.


**Limitations And Societal Impact:**

The authors discuss the scaling limitations of their work as they related to the cubic scaling with the number of atoms; there are no potential negative societal impacts of this work.

**Main Review:**


The method is a clear improvement over SchNOrb: it incorporates SO3 symmetries in its core and it shows significantly better approximation accuracy, better data efficiency, and reduced model size. The paper is written in a clear and concise way, though the demonstrations are somewhat lacking compared to what I'd have expected for the NeurIPS conference. The possible combinations of the building blocks all make sense, but I'd have liked to see an ablation study or additional arguments in the supplement for the particular combinations of architectural choices. Importantly, the authors do not discuss how/if this architecture could generalize to predict results on a system not seen in the training set. They also don't show performance on total energy when using the previously established 1K MD17 training datasets, so it's unclear how much this contribution adds to practical applications. Although the prediction of DFT level wavefunctions for new molecular conformations is potentially interesting, I was more excited about the possibility to correct these wavefunction to an effective CCSD(T) wavefunction. Unfortunately, the example with a single illustration of a single water molecule is not very convincing that this architecture can provide useful practical tool in the near future.

Edit: following the additional work that demonstrates the performance gains on DFT-level calculations, I've updated my overall score to 6.


**Time Spent Reviewing:**

10

---

> ### Author Response · Authors · 2021-08-09
> **Response to Reviewer 5CND**
>
> We would like to clarify that the compute costs of our method are significantly lower than those of Hartree Fock calculations, because Eq. 3 needs to be solved only once with predicted matrices (our method), instead of repeatedly solving Eq. 3 until a self-consistent solution is found (Hartree Fock).
>
> Thank you for suggesting ablation studies to motivate the combination of architectural choices, which we will include in the supporting information for the camera-ready version. In particular, we tested the following four ablated variants of our model:
> - shared matrix features: The same features are used for central and neighboring atoms for the Hamiltonian matrix prediction (see Eq. 16, errors increase between 6-30%).
> - shared interaction features: The same features are used for central and neighboring atoms in interaction blocks (see Eq. 13, errors increase between 16-54%).
> - no selfmix: All selfmix (Eq. 9) operations are removed from spherical linear layers (Eq. 10) when the input and output degree $L$ is equal (errors increase between 14-77%).
> - simple pair features: No interaction with neighboring atoms is used for constructing self-interaction (Eq. 17) and pair-interaction (Eq. 18) features (errors increase between 57-360%).
>
> Further details and a discussion of these results will be added to the supporting information.
>
> We agree that it would be highly interesting to test the generalization capabilities of our model across different chemical compounds. Note however, this would require the generation of a large dataset of Hamiltonian matrices for different chemical compounds (in different configurations), which is a major computing effort. While such a benchmark (which does not exist yet) would definitely be of value for the community, its generation goes beyond the scope of the present work and in our opinion merits a separate publication.
>
> Unfortunately, it is unfeasible to compare the energy predictions of our model to the MD17 dataset benchmarks as is.
> The MD17 benchmark is intended for machine learning models which directly predict energies and forces (instead of the wavefunction). The included datasets therefore do not contain reference data for Hamiltonian matrices, which our model requires for training. Evaluating our model on the MD17 datasets would require re-calculating every data point to obtain the necessary reference data, which requires major computational resources beyond the scope of the present work.
> Using the current models trained on subsets of MD17 in direct comparisons is problematic as well, since the reference data uses a different computational method (Gaussian basis sets instead of numerical ones) than the original MD17.
>
> We want to emphasize that the practical relevance of our model lies in its ability to accurately predict molecular wavefunctions, from which a large number of other quantities of interest can be derived (e.g. orbital energies and populations). In applications where only energies and forces are required, we agree that simpler models are a more appropriate choice.
> The ability of our model to correct wavefunctions to an effective CCSD(T) wavefunction was shown as a proof of principle and will be explored further in future work.
> Given the accuracy achieved for the aspirin wavefunction and the fact that CCSD(T) corrections are small compared to the HF wavefunction (the recovered correlation energy is small compared to the total electronic energy, see the water example), we are positive that our findings will hold for more practical systems.
> The main difficulty will be generating the reference computations, as the cost of CCSD(T) rises rapidly with increased molecule size.
> This was also the main reason why we restricted ourselves to a small molecule where extensive testing was still feasible without a major computing effort.
>
> However, even though you may find this particular example unconvincing, we hope that we can convince you that the prediction of DFT level wavefunctions alone is very relevant in practice. Our model can be used to entirely skip the expensive iterations in traditional DFT calculations required to obtain a self-consistent solution. Alternatively, its predictions can be used as a sophisticated initial guess (much closer to the true solution than typical guess wavefunctions), thus achieving a self-consistent solution with fewer iterations. This way, traditional DFT calculations can be accelerated without any trade-offs in accuracy or robustness, which makes our method attractive even for practitioners who might mistrust pure machine learning solutions.
>
> If you think the additions we want to include in the camera-ready version (e.g. ablation studies) improve the quality of our work, we kindly ask you to consider increasing your rating.

---

> > ### Comment · Reviewer_5CND · 2021-09-02
> > **I remain unconvinced, unfortunately.**
> >
> > Thank you for the additional ablation studies and for clarifying the difference in performance compared to Hartree Fock calculations.  I am still unconvinced that this method contributes to the state of the art, since, for example, the energy error in the Aspirin calculation, although impressively low, is still double the comparable error listed in the preprint (and submission to NeurIPS) from the method called UNiTE.  I also don't fully understand the hesitation for creating QM data for part of the MD17 dataset: I thought that the computational cost for getting such data is comparable to Hartree Fock calculations and one probably only needs 1000 of them (but maybe I'm still missing something).  I could have been convinced about the importance of this method from a benchmark documenting the performance gains in using the new method as a quick way to converge DFT calculations (or even HF calculations for that matter), but the authors only mentioned this concept and didn't support it with any practical and complete examples.  I still remain skeptical about the possible generalization of this method to novel molecules (unseen in the training set) but I agree with the authors that if they achieve to do such a demonstration it would deserve a publication on its own and such a publication would have a chance to be in the top 50% of neurIPS.  As it stands, I remain with my original score of 4 for the present paper.

---

> > > ### Author Response · Authors · 2021-09-06
> > > **Clarifications and new experiments for improving the convergence of DFT calculations**
> > >
> > > We would like to reiterate that the method we present here aims to predict molecular wavefunctions and electronic densities, which is a different (and harder) problem than directly predicting energies. We target these quantities, as they provide the full ground state electronic structure information of a molecule.
> > > This makes it possible to compute a wide range of properties by applying well established expressions from quantum mechanics.
> > > Energies and forces are merely chosen as an example of properties that can be derived from the wavefunction, however, their prediction is not the main focus of our work.
> > > In applications where only energies are required, simpler models that directly predict energies are preferable, as the "detour" over the wavefunction would be an unnecessary complication. As such, focusing solely on energy predictions (and comparisons to specialized models for this purpose) is generally not the best way to appreciate the fundamental properties of our (wavefunction) method. While it is true (and impressive) that UNiTE achieves low energy prediction errors, we would, however, like to point out that this method is similar to OrbNet in that it requires orbital features as inputs (see our response to Reviewer ZUAT). Additionally, UNiTE follows a delta-learning approach, i.e. it only predicts a correction, which is added to a semi-empirical baseline, to obtain its energy prediction. Finally, UNiTE is evaluated on the "revised MD17" data set, which uses a finer DFT grid for the reference calculations. This was shown to simplify energy predictions due to lower numerical noise in the reference data (see https://doi.org/10.1088/2632-2153/abba6f) and makes direct comparisons with benchmarks performed on the original MD17 data set somewhat problematic.  While you are correct that models are typically only trained on 1000 points for the MD17 benchmark, we believe a fair comparison with published benchmark results would require evaluation under exactly the same conditions, i.e. also using the same test set. This would require over 3.6M calculations to cover all molecules in the MD17 benchmark (see http://quantum-machine.org/gdml/#datasets). Additionally, for direct comparisons to UNiTE, we would even need to perform additional computations with the DFT grid settings of the revised MD17 dataset (https://doi.org/10.1088/2632-2153/abba6f). So the suggested comparison not only goes well beyond the present submission but asks for aspects peripheral to the punchline of our manuscript, which is introducing a novel powerful wavefunction method.
> > >
> > > We thank you very much for your highly constructive remark, namely for suggesting a benchmark with practical and complete examples for improving the convergence of DFT/HF calculations with our method! We performed these experiments and have found that when starting DFT calculations with a wavefunction predicted by our model as an initial guess, the number of iterations required for finding a self-consistent solution decreases on average between 56-72% (depending on the molecule). This translates to an average total reduction in wall-clock time between 24-40%, without any loss in accuracy. Detailed results will be included in the camera-ready version and we hope that this will convince readers that remain skeptical of the importance of our contribution.

---

> > > > ### Comment · Reviewer_5CND · 2021-09-10
> > > > **Thank you for the additional demonstration**
> > > >
> > > > With the addition of the ablation studies and, importantly, the demonstration of a reduction in the time to solution for DFT calculations, the paper has improved significantly and passes the bar for acceptance at this conference.  I've updated my score accordingly.

---

### Official Review · Reviewer_jstg · 2021-07-16

**Rating:** 7
**Confidence:** 2

**Summary:**

This paper describes PhiSNet, a model for predicting the Hamiltonians of systems of atoms in the context of quantum chemistry. The output Hamiltonian can then be used to derive the wavefunction using traditional methods, and the wavefunction can in turn be used to derive physically observable properties of interest. A core part of the design of this model is built-in equivariance to rotation in three dimensions. The paper defines the architecture of the PhiSNet and each of its equivariant component layers in detail, and results are presented from a comparison to the competing method SchNOrb.

**Limitations And Societal Impact:**

The authors adequately address some limitations of their work in section 5. They do not discuss potential societal impact - it is not clear to me that such a discussion is necessary for this paper.

**Main Review:**

In general the paper is well-written and well-structured, with about the right amount of space devoted to each part (e.g. the background in section 3 is pretty minimal, but I think this is the correct decision here, and I appreciated the longer-form background provided in the supplement). As with almost all papers, it would benefit from another proofreading pass to catch minor errors (e.g. "calculations and led to" on line 25 should be "calculations, and have led to"). The detailed description of PhiSNet and its component parts is clear and well laid out. Figure 1 is excellent, I found it extremely helpful in clarifying the problem setting and the architecture.

There are, as the authors note in section 2, a number of existing SE(3)-equivariant machine learning methods. In section 4.1, it is unclear which of the building blocks the authors are claiming as novel and which are taken from (or heavily based on) these earlier works. They say "we describe novel general-purpose operations" on line 118-119, but clearly not everything that follows is novel (e.g. the chosen activation function and the linear layers). I would appreciate more clarity on this.

A point about terminology: I'm unclear exactly what the authors mean by "irrep". Strictly an irrep is a representation, i.e. a homomorphism from SO(3) to GL(N) (which satisfies some additional properties to be "irreducible"), but the authors also seem to refer to feature vectors as "irreps" (e.g. in section 4 on line 152). Is the idea that SO(3) acts on these feature vectors by an irrep?

The results appear impressive, though since I am unfamiliar with this application domain I am taking it on good faith that the results in Table 1 represent a fair comparison against a state-of-the-art competitor. The results are the most compelling part of the paper, and they are the main thing that convinces me of the value of the model architecture described in section 4.

**Time Spent Reviewing:**

4 hours

---

> ### Author Response · Authors · 2021-08-09
> **Response to Reviewer jstg**
>
> Thank you for the constructive feedback, we made further proofreading passes through the paper and will correct the errors you pointed out in the camera-ready version.
>
> You are right that not all components are novel and some are based on previous work in this area. The tensor product expansion is used for the first time in the context of SE(3)-equvariant networks here, and while other components may be similar to previous work, we believe the way individual building blocks are combined in the model architecture to be unique and a major factor to the accuracy and versatility of our model.
> We would also like to note that the choice of potential building blocks is inherently limited due to the equivariance constraints, leading to the use of some already established components.
> In this context, we chose to discuss components like activation functions and linear layers, because even though they are not novel, they have to be used in specific ways to keep predictions rotationally equivariant, which may not be obvious to all readers.
> We are happy to further clarify and revise the manuscript accordingly.
>
>
> The paragraph including line 152 was meant to be a more general description that does not only apply to features in the context of a neural network. Tensor product contractions allow to take the tensor product of two irreducible representations (which is reducible), and express it again with an irredecuible representation. We will revise this paragraph to make the meaning of "irreps" and "feature vectors" more clear.

---

> > ### Comment · Reviewer_jstg · 2021-08-19
> > **Response to rebuttal**
> >
> > Thanks for confirming that you can include some revisions on the points raised in my review. My rating remains at 7.

---

### Decision · Program_Chairs · 2021-09-27

**Decision:**

Accept (Poster)

**Comment:**

The authors present an SE(3) equivariant architecture for predicting symmetric Hamiltonian matrices, which can then be used to derive physical properties of molecules. This architecture provides a dramatic improvement in accuracy over baseline methods. The majority of reviewers agreed it was an excellent paper deserving acceptance. The one holdout reviewer had concerns about the scale of the experiments and recommended that results be presented for the entirety of the MD17 dataset. However I am convinced by the author response that this would not be feasible with the computational resources at their disposal, and I am willing to cut them some slack on this given the quality of results on the systems presented. Therefore I recommend that the paper be accepted.